# MAXCUTPOOL: DIFFERENTIABLE FEATURE-AWARE MAXCUT FOR POOLING IN GRAPH NEURAL NETWORKS

**Carlo Abate** *
Alma Mater Studiorum - University of Bologna
Fondazione Istituto Italiano di Tecnologia
`carlo.abate@iit.it`

**Filippo Maria Bianchi** *
UiT the Arctic University of Norway
NORCE Norwegian Research Centre AS
`filippo.m.bianchi@uit.no`

## ABSTRACT

We propose a novel approach to compute the MAXCUT in attributed graphs, *i.e.*, graphs with features associated with nodes and edges. Our approach works well on any kind of graph topology and can find solutions that jointly optimize the MAXCUT along with other objectives. Based on the obtained MAXCUT partition, we implement a hierarchical graph pooling layer for Graph Neural Networks, which is sparse, trainable end-to-end, and particularly suitable for downstream tasks on heterophilic graphs.

## 1 INTRODUCTION

The MAXCUT is the problem of partitioning the nodes of a graph such that as many edges as possible connect nodes from different sides of the partition. The MAXCUT is orthogonal to the more commonly encountered minCUT, which aims at partitioning the nodes into strongly connected groups. While minCUT is closely related to clustering, MAXCUT relates to the concept of downsampling, *e.g.*, keeping one-every-$K$, under the assumption that there is a redundancy among the $K$ samples. Like the minCUT, the MAXCUT is a combinatorial optimization problem that, in practice, is approximated by approaches that find suboptimal or unstable solutions for a large class of graphs (Makarychev et al., 2014).

Pooling is ubiquitously used in deep learning for gradually reducing the size of the data while retaining important information. In Convolutional Neural Networks (CNNs), pooling is typically implemented by selecting the maximum within a contiguous patch (*max-pool*) or by computing an average (*avg-pool*). These strategies are naturally related to MAXCUT and minCUT problems, respectively. Similarly to CNNs, Graph Neural Networks (GNNs), which can be seen as a generalization to irregular data, can be built by alternating Message Passing (MP) and graph pooling layers (Zhou et al., 2020a). A hierarchy of pooling layers gradually extracts global graph properties through the computation of local summaries and is key to building deep GNNs for graph classification (Khasahmadi et al., 2020), node classification (Gao & Ji, 2019; Ma et al., 2020), graph matching (Liu et al., 2021), and spatio-temporal forcasting (Cini et al., 2024; Marisca et al., 2024).

Two important approaches are followed when implementing hierarchical graph pooling. One is to account for the node features with trainable functions that are adapted to a downstream task at hand. The other is to optimize graph theoretical objectives, such as the minCUT or the MAXCUT, to guide the computation of the coarsened graph. Combining the first approach with minCUT objectives is relatively straightforward, as they complement the smoothing effect of MP layers (Hansen & Bianchi, 2023). Conversely, objectives such as MAXCUT that select sparse and uniformly distributed subsamples of nodes have been implemented so far only within non-differentiable frameworks, which account neither for node features nor for task objectives (Luzhnica et al., 2019).

### 1.1 CONTRIBUTIONS

**MAXCUT for attributed graphs.** Our first contribution is graph theoretical and consists of a novel GNN-based approach to compute a MAXCUT partition in attributed graphs. Being differentiable, our

---
*Equal contribution

method can be seamlessly integrated into a deep-learning framework where other loss functions can influence the MAXCUT solution. Remarkably, our method is also more robust than traditional approaches in computing the MAXCUT on non-attributed graphs, as it finds a better cut on most graph topologies. This makes our contribution relevant to *every* application of the MAXCUT problem, such as quantum computing (Zhou et al., 2020b), circuit design (Bashar et al., 2020), statistical physics (Borgs et al., 2012), material science (Liers et al., 2004), computer vision (Abbas & Swoboda, 2022), and quantitative finance (Lee & Constantinides, 2023).

**Graph pooling and coarsening.**    The MAXCUT application we focus on is the problem of learning a coarsened graph within a GNN. In particular, we design a new hierarchical pooling layer that reduces the graph by keeping the nodes from one side of the MAXCUT partition. Our layer is the first to combine a graph theoretical MAXCUT objective with a pooling approach that is features-aware and trainable end-to-end. When we include the newly proposed pooling layer in GNNs for graph and node classification, we achieve similar or superior performances compared to state-of-the-art pooling techniques.

**Improved scoring-based pooling framework.**    We propose a simple and efficient scheme to assign nodes to supernodes when computing the pooled graph. Our scheme can be applied not only to our method but to the whole family of sparse scoring-based graph pooling operators enhancing, in principle, their representational power. Importantly, we bridge the gap between scoring-based and dense pooling methods by using the same operations to compute the features and the topology of the pooled graph.

**Heterophilic graph classification dataset.**    Differently from the existing differentiable pooling operators, the nature of the MAXCUT solution makes our graph pooling operator particularly suitable for heterophilic graphs. While there are benchmark datasets for node classification on heterophilic graphs, there is a lack of such datasets for graph classification. To fill this gap, we introduce a novel synthetic dataset that, to our knowledge, is the first of its kind.

## 2 BACKGROUND

### 2.1 THE MAXCUT PROBLEM AND THE CONTINUOUS RELAXATIONS

Let $\mathcal{G} = (\mathcal{V}, \mathcal{E})$ be an undirected graph with non-negative weights on the edges, and let $N$ be the number of nodes in $\mathcal{G}$. A cut in $\mathcal{G}$ is a partition $(\mathcal{S}, \mathcal{V} \setminus \mathcal{S})$ where $\mathcal{S} \subset \mathcal{V}$. The MAXCUT problem consists of finding a cut that maximizes the total volume of edges connecting nodes in $\mathcal{S}$ with those in $\mathcal{V} \setminus \mathcal{S}$. The MAXCUT objective can be expressed as the integer quadratic problem

$$\max_{\boldsymbol{z}} \sum_{i,j \in \mathcal{V}} w_{ij}(1 - z_i z_j) \ \text{ s.t. } \ z_i \in \{-1, 1\}, \tag{1}$$

where $\boldsymbol{z} \in \{-1, 1\}^N$ is an assignment vector indicating to which side of the partition each node is assigned to and $w_{ij}$ is the weight of the edge connecting nodes $i$ and $j$.

Like other discrete optimization problems of this kind, MAXCUT is NP-hard. The Goeman-Williamson (GW) algorithm (Goemans & Williamson, 1995) provides a semidefinite relaxation of the integer quadratic problem, which makes it tractable:

$$\max_{\boldsymbol{X}} \sum_{i,j \in \mathcal{V}} w_{ij}(1 - \boldsymbol{x}_i \cdot \boldsymbol{x}_j) \ \text{ s.t. } \ \|\boldsymbol{x}_i\| = 1, \tag{2}$$

where $\boldsymbol{X} \in \mathbb{R}^{N \times D}$ is a matrix whose rows are the continuous embeddings of the nodes in $\mathcal{G}$. The vectors $\boldsymbol{X}$ are projected on a random hyperplane to split the nodes and assign them to the two sides of the partition. This algorithm guarantees an expected cut size of .868 of the maximum cut.

Another simple yet effective continuous relaxation is the largest eigenvector vertex selection (LEVS) method (Shuman et al., 2015). Let $\boldsymbol{L}$ be the Laplacian matrix associated to the graph $\mathcal{G}$ and let $\boldsymbol{u}_{\max}$ be the eigenvector of $\boldsymbol{L}$ associated to the largest eigenvalue $\lambda_{\max}$. A cut in $\mathcal{G}$ can be found based on the polarity of the components of $\boldsymbol{u}_{\max}$, for instance by letting $\mathcal{S} = \{i : \boldsymbol{u}_{\max}[i] \geq 0\}$. In the field of graph signal processing, the eigenvectors related to the largest eigenvalues of $\boldsymbol{L}$ are closely related to

the operation of high-pass filtering of a graph signal (Tremblay et al., 2018). Specifically, they are used to design graph filters that amplify *high-frequency* components of a signal, *i.e.*, the components that vary the most across adjacent nodes (Shuman et al., 2013).

The MAXCUT problem is closely related to *graph coloring*, which aims at assigning different colors to adjacent nodes. In particular, the 2-colors *approximate coloring* (Aspvall & Gilbert, 1984) is the problem of identifying subsets of nodes such that the connections within each subset are minimized. Such coloring is a high-frequency graph signal and induces a partition that is orthogonal to spectral clustering (von Luxburg, 2007).

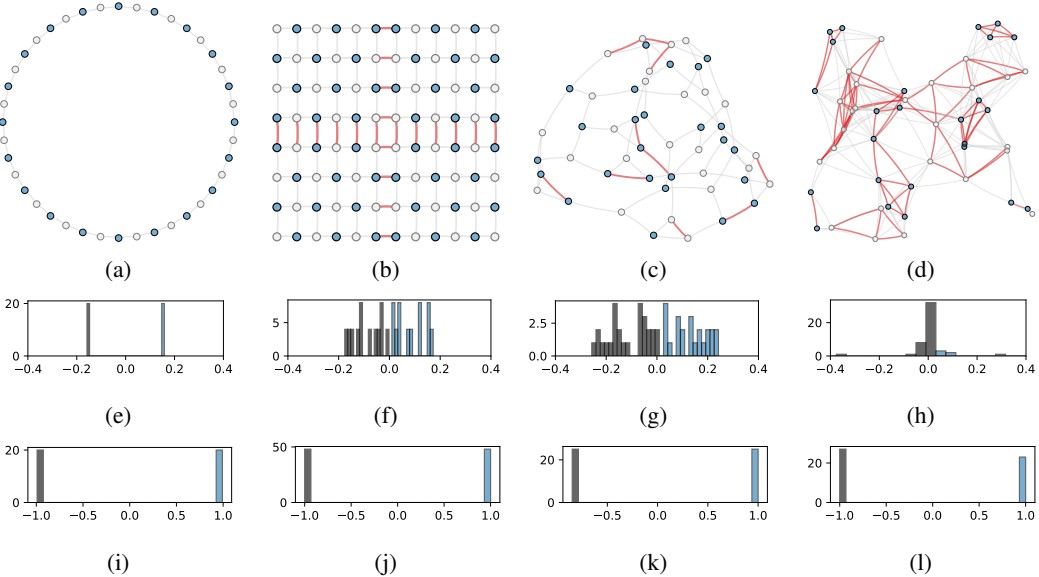

Figure 1: **Top row:** Partitions induced by the sign of the elements in $\boldsymbol{u}_{\max}$. The nodes are colored based on the partition, and the red edges are those not cut (the fewer, the better). **Middle row:** histograms of $\boldsymbol{u}_{\max}$ inducing the partitions above. While in bipartite graphs the separation is sharp, the more a graph is irregular and dense, the more the values are clustered around zero, making it difficult to find the optimal MAXCUT. **Bottom row:** histograms of the score vectors generated by our model, which always produce a clear and sharp partition.

A MAXCUT partition that cuts every edge exists only for bipartite graphs. Conversely, in fully connected graphs, no more than half of the edges can be cut. Algorithms relying on continuous relaxations to find the MAXCUT partition tend to be unstable and perform poorly, especially when the graph topology departs from the bipartite case (Trevisan, 2009). Fig. 1(a-h) shows the performance of the LEVS method on bipartite and non-bipartite graphs: numerical issues typically occur in more dense and less regular graphs, making it difficult to identify the optimal MAXCUT solution.

## 2.2 Message passing in GNNs

Let us consider a graph $\mathcal{G} = \left(\boldsymbol{X} \in \mathbb{R}^{N \times F}, \boldsymbol{A} \in \mathbb{R}^{N \times N}\right)$. A basic MP operator can be described as

$$\boldsymbol{X}' = \sigma\left(\boldsymbol{P}\boldsymbol{X}\boldsymbol{\Theta}\right) \tag{3}$$

where $\sigma$ is a non linear activation function, $\boldsymbol{\Theta}$ are trainable parameters, and $\boldsymbol{P}$ is a propagation operator matching the sparsity pattern of $\boldsymbol{A}$. Each MP layer relies on a specific propagation operator. For instance, in Graph Convolutional Networks (GCNs) (Kipf & Welling, 2017), the propagation operator is defined as $\boldsymbol{P} = \hat{\boldsymbol{D}}^{-\frac{1}{2}}\hat{\boldsymbol{A}}\hat{\boldsymbol{D}}^{-\frac{1}{2}}$, where $\hat{\boldsymbol{A}} = \boldsymbol{A} + \boldsymbol{I}$ and $\hat{D}_{ii} = \sum_{j=0}\hat{A}_{ij}$.

Due to the fixed, non-negative smoothing nature of common propagation operators, the repeated application of $\boldsymbol{P}$ can lead to over-smoothing. If that happens, the feature representations of nodes become increasingly similar, hindering the function approximation capabilities of a GNN, which can only learn smooth graph signals (Wu et al., 2019; Wang et al., 2019). In contrast, by combining smoothing propagation operators with *sharpening* ones, any kind of gradients can be learned (Eliasof

et al., 2023; Bianchi et al., 2021). Since there is no universal definition for such an operator, we rely on the formulation introduced by Bianchi (2022):

$$P = I - \delta \left( I - D^{\frac{1}{2}} A D^{\frac{1}{2}} \right) = I - \delta L^{\text{sym}} \tag{4}$$

where $\delta$ is a smoothness hyperparameter and $L^{\text{sym}}$ is the symmetrically normalized Laplacian of $\mathcal{G}$. As observed by Bianchi (2022), when $\delta = 0$ the MP behaves like a simple Multilayer Perceptron (MLP). Instead, when $\delta = 1$, the behavior is close to that of a GCN. Finally, as noted by Eliasof et al. (2023), when $\delta > 1$ the propagation operator favors the realization of non-smooth signals on the graph, and we refer to this variant as a Heterophilic Message Passing (HetMP) operator. We note that this can be seen as the graph counterpart of the Laplacian sharpening kernels for images, mapping connected nodes to different values (Mather & Koch, 2022).

## 2.3 Graph pooling

While there are profound differences between existing graph pooling approaches, most of them can be expressed through the Select-Reduce-Connect (SRC) framework (Grattarola et al., 2022). Specifically, a pooling operator $\texttt{POOL} : (A, X) \mapsto (A', X')$ can be expressed as the combination of three sub-operators:

- $\texttt{SEL} : (A, X) \mapsto S \in \mathbb{R}^{N \times K}$, is a selection operator that defines how the $N$ original nodes are mapped to the $K$ pooled nodes, called *supernodes*, being $S$ the *selection matrix*.

- $\texttt{RED} : (X, S) \mapsto X' \in \mathbb{R}^{K \times F}$, is a reduction operator that yields the features of the supernodes. A common way to implement $\texttt{RED}$ is $X' = S^\top X$.

- $\texttt{CON} : (A, S) \mapsto A' \in \mathbb{R}^{K \times K}_{\geq 0}$, is a connection operator that generates the new adjacency matrix and, potentially, edge features. Typically, $\texttt{CON}$ is implemented as $A' = S^\top A S$ or $A' = S^+ A S$.

Different design choices for $\texttt{SEL}$, $\texttt{RED}$, and $\texttt{CON}$ induce a taxonomy of the operators. For example, if any of $\texttt{SEL}$, $\texttt{RED}$, and $\texttt{CON}$ is learned end-to-end, the pooling operators are called *trainable*, *non-trainable* otherwise. Relevant to this work are the families of pooling methods described below.

**Soft-clustering** methods, sometimes referred to as *dense* (Grattarola et al., 2022), assign each node to more than one supernode through a soft membership. Representatives methods such as DiffPool (Ying et al., 2018), MinCutPool (Bianchi et al., 2020a), StructPool (Yuan & Ji, 2020), HoscPool (Duval & Malliaros, 2022), and Deep Modularity Networks (DMoN) (Tsitsulin et al., 2023), compute a soft cluster assignment matrix $S \in \mathbb{R}^{N \times K}$ either with an MLP or an MP-layer operating on the node features and followed by a $\texttt{softmax}$. Most of these methods are trainable and leverage unsupervised auxiliary loss functions to guide the formation of the clusters. Soft-clustering methods usually perform well on downstream tasks due to their flexibility and *expressive power*, which is the capability of retaining all the information from the original graph (Bianchi & Lachi, 2023). However, storing the soft assignments $S$ is a memory bottleneck for large graphs (see, *e.g.*, the analysis of memory usage in Appendix F.3), and soft memberships make pooled graphs very dense and not interpretable. Additionally, each graph is mapped to the same fixed number of supernodes $K$, which can hinder the generalization capabilities in datasets where the size of each graph varies significantly.

**Scoring-based** methods select supernodes from the original nodes based on a node scoring vector $s$. The chosen nodes correspond to the top $K$ elements of $s$, where $K$ can be a ratio of the nodes in each graph, making these methods adaptive to the graph size. Representatives such as Top-$k$ Pooling (Top-$k$) (Gao & Ji, 2019; Knyazev et al., 2019), ASAPool (Ranjan et al., 2020), SAGPool (Lee et al., 2019), PanPool (Ma et al., 2020), TAPool (Gao et al., 2021), CGIPool (Pang et al., 2021), and IPool (Gao et al., 2022) primarily differ in how they compute the scores or in the auxiliary tasks they optimize to improve the quality of the pooled graph. Despite a few attempts to encourage diversity among the selected nodes (Zhang et al., 2019; Noutahi et al., 2019), scoring-based methods derive the scores from node features that tend to be locally similar, especially after being transformed by MP operations. As such, the pooled graph often consists of a chunk of strongly connected nodes with similar characteristics. Consequently, entire sections of the graph are not represented, reducing the expressiveness and lowering performance in downstream tasks (Wang et al., 2024).

**One-every-$K$** methods leverage graph-theoretical properties to select supernodes by subsampling the graph uniformly. For instance, $k$ Maximal Independent Sets Pooling ($k$-MIS) (Bacciu et al., 2023)

identifies as supernodes the members of a maximal $K$-independent set, *i.e.*, nodes separated by at least $K$-hops on the graph. Graclus (Dhillon et al., 2007; Defferrard et al., 2016) creates supernodes by merging the pairs of most connected nodes in the graph. SEP (Wu et al., 2022) partitions the node hierarchically according to a precomputed tree that minimizes the structural entropy of the graph. Node Decimation Pooling (NDP) (Bianchi et al., 2020b) divides the graph into two sets, $\mathcal{V}_+$ and $\mathcal{V}_-$, according to the partition induced by the components of $\boldsymbol{u}_{\max}$ (see Section 2.1). One of the two sides of the partition is dropped ($\mathcal{V}_-$), while the other ($\mathcal{V}_+$) becomes the set of supernodes. While both Graclus and NDP can only reduce the number of nodes by approximately half, higher pooling ratios (one-every-$2^K$) are achieved by applying them recursively $K$ times. Nevertheless, they lack the same control of soft-clustering and scoring-based methods in fixing the size of the pooled graphs. Like the scoring-based methods, one-every-$K$ methods are adaptive and produce crisp cluster assignments. However, they are not trainable and precompute the pooled graph based on the topology without accounting for the node features or the downstream task. Tab. 1 summarizes the drawbacks of the existing families of pooling methods.

Table 1: Drawbacks of different types of pooling operators.

| Soft-clustering | Score-based | One-every-$K$ |
|---|---|---|
| ✗ Not adaptive to graph size | ✗ Pooling not uniform | ✗ Limited flexibility |
| ✗ Dense and not interpretable pooled graphs | ✗ Not expressive | ✗ Features agnostic |
| ✗ High memory cost | ✗ Worse performance | ✗ Task agnostic |

## 3 METHOD

We leverage a GNN to generate a MAXCUT partition while accounting for node features and additional objectives from a downstream task. In particular, we let node features and task-specific losses influence the MAXCUT solution, creating partitions that not only maximize the number of cut edges but also prioritize the selection of nodes that are optimal for the downstream task at hand. To reach this goal, it is necessary to overcome a tension between the effect of a standard MP layer and the MAXCUT: the former applies a smoothing operation that makes adjacent nodes similar, which is orthogonal to the objective of the latter. Therefore, to implement MAXCUT with a GNN we rely on HetMP, implemented by setting $\delta > 1$ in the MP operation in Eq. 4. As discussed in Sec. 2.1, solving the MAXCUT problem is equivalent to coloring adjacent nodes differently. Notably, this is an intrinsic effect of HetMP that makes features of adjacent nodes as different as possible, effectively acting as a high-pass graph filter. Therefore, optimizing a MAXCUT loss on features generated by HetMP layers overcomes the limitation of traditional scoring-based methods that compute the scores from features produced by homogeneous MP operators.

The layer we propose is called MaxCutPool and we present it through the SRC framework. The SEL operation in MaxCutPool identifies as supernodes a subset $\mathcal{S}$ of the nodes in the original graph. An auxiliary GNN, called *ScoreNet*, consists of a stack of HetMP layers that map the node features into a vector $\boldsymbol{s} = \text{ScoreNet}(\boldsymbol{X}, \boldsymbol{A}) \in [-1, 1]^N$, which assigns a score to each node. The indices $\boldsymbol{i} = \text{top}_K(\boldsymbol{s})$ associated with the highest scores identify the $K$ supernodes. Additional details about the ScoreNet are in Appendix C.1. Fig. 1(i-l) shows the histograms of the score vectors $\boldsymbol{s}$ generated by the ScoreNet for the 4 example graphs. Compared to the histograms of $\boldsymbol{u}_{\max}$ in Fig. 1(e-h), the values in $\boldsymbol{s}$ always produce a distribution with two sharp and well-separated modes, yielding a clear node partition.

After the $K$ supernodes are selected, the remaining $N - K$ nodes are assigned to one of the supernodes via the nearest neighbor aggregation. An assignment matrix $\boldsymbol{S}$ is built by performing a breadth-first visit of the graph where, starting from the supernodes, all the remaining nodes are assigned to their nearest supernode (see Fig. 2). More formally, the assignment matrix $\boldsymbol{S}$ is defined as

$$\text{SEL} : [\boldsymbol{S}]_{ij} = 1 \iff j = \phi(\mathcal{S}, \boldsymbol{A}, i),$$

where $\phi(\mathcal{S}, \boldsymbol{A}, i)$ returns the closest supernode to node $i$. The visit of the graph is iterated until all nodes are assigned to a supernode or until a maximum number of iterations is reached. In the latter case, nodes that are still unassigned are assigned at random to ensure that the pooled graph is always connected. Keeping the maximum number of iterations small (*e.g.*, 2 or 3) prevents

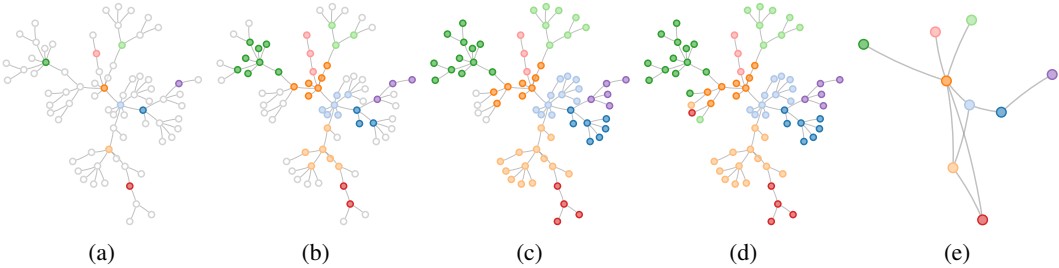

Figure 2: **(a)** The nodes with the $K = 9$ highest scores are selected. **(b-c)** Their ID is propagated to the unselected nodes until all are covered or until a maximum number of iterations (2 here) is reached. **(d)** The 4 remaining nodes are assigned randomly. **(e)** The pooled graph is obtained by aggregating the nodes with the same ID and coalescing the edges connecting nodes from different groups.

pointless attempts to reach a supernode, *e.g.*, when there are no supernodes within a disconnected graph component, and also injects randomness acting as a regularizer that helps to move away from suboptimal configurations often encountered at the beginning of the training stage. The pseudo-code for a GPU-parallel implementation of the proposed assignment scheme is deferred to Appendix A.

The RED operation for computing the features of the supernodes can be implemented in two ways:

$$\text{RED: } \boldsymbol{X}' = s_i \odot [\boldsymbol{X}]_i \quad \text{or} \quad \text{RED: } \boldsymbol{X}' = \boldsymbol{s_i} \odot \boldsymbol{S}^\top \boldsymbol{X}.$$

The Hadamard product $\odot$ enables gradients flowing through the ScoreNet during back-propagation, making it possible for the gradients of the task loss to reach every component of our model, despite the non-differentiable top$_K$ operation. When the first or the second variant is used to implement RED, we refer to the pooling operator as MaxCutPool and MaxCutPool-E, respectively. The suffix "-E" indicates that the pooling operator satisfies the sufficient conditions for expressiveness defined by Bianchi & Lachi (2023).

Finally, the CON operation can be implemented as

$$\text{CON} : \boldsymbol{A}' = \boldsymbol{S}^\top \boldsymbol{A} \boldsymbol{S}.$$

### 3.1 AUXILIARY LOSS

Each MaxCutPool layer is associated with an auxiliary loss that encourages the top-$K$ selected nodes to belong to the same side of the MAXCUT partition. The loss is defined as:

$$\mathcal{L}_{\text{cut}} = \frac{\boldsymbol{s}^\top \boldsymbol{A} \boldsymbol{s}}{|\mathcal{E}|} \tag{5}$$

where $|\mathcal{E}| = \sum_{ij} w_{ij}$ is the total edge weight of the graph. Since $-1 \leq s_i \leq 1$, we have $-|\mathcal{E}| \leq \boldsymbol{s}^\top \boldsymbol{A} \boldsymbol{s} \leq |\mathcal{E}|$, hence $-1 \leq \mathcal{L}_{\text{cut}} \leq 1$. This loss evaluates the ratio between the volume of the cut and the total volume of the edges. Minimizing $\mathcal{L}_{\text{cut}}$ encourages the nodes to be assigned to different partitions if and only if they are connected. The loss reaches its minimum $-1$ when all connected nodes are assigned to opposite sides of the partition, *i.e.*, when all the edges are cut. Clearly, this can happen only in bipartite graphs. The details about the derivation of the loss are in Appendix B.

A GNN model consisting of MP layers interleaved with MaxCutPool layers can be trained end-to-end to jointly minimize a task loss $\mathcal{L}_{\text{task}}$ and the auxiliary loss $\mathcal{L}_{\text{cut}}$. The total loss is then defined as

$$\mathcal{L} = \mathcal{L}_{\text{task}} + \sum_l \beta \mathcal{L}_{\text{cut}}^{(l)} \tag{6}$$

where $\beta$ is a scalar weighting each auxiliary loss $\mathcal{L}_{\text{cut}}^{(l)}$ associated with the $l$-th MaxCutPool layer.

### 3.2 HOMOPHILIC AND HETEROPHILIC OPERATIONS

Despite the presence of HetMP layers and the heterophilic loss, MaxCutPool can be inserted into GNNs equipped with traditional MP layers. In fact, our method leverages the homophilic nature of

the latter. After a standard homophilic MP, the stronger the association between a pair of nodes, the more similar their features will be. Keeping them both will, thus, be redundant, and one of them can be dropped. This is precisely what is done by SEL that, thanks to its heterophilic design, samples nodes that share as few connections as possible and are uniformly distributed over the graph.

MaxCutPool contains an additional homophilic operation: the computation of the assignments $S$ through the nearest-neighbor aggregation, which synergizes with the repulsiveness in the supernodes' sampling. The assignments are used by CON to produce a connectivity matrix that is connected yet sparse, and they can also be leveraged by RED to ensure the expressiveness of the pooling layer. However, in heterophilic datasets where such a homophilic assignment is not ideal, the non-expressive variant of MaxCutPool offers a more suitable alternative.

Overall, rather than creating tension, combining homophilic and heterophilic components provides a GNN with MaxCutPool the flexibility for handling different scenarios. As we show in the experimental evaluation, our model can even switch to a completely homophilic setting by adjusting the values of $\delta$ and $\beta$ or by ignoring the auxiliary loss $\mathcal{L}_{\text{cut}}$.

### 3.3 RELATION WITH OTHER POOLING METHODS

MaxCutPool belongs to the scoring-based pooling family from which it inherits the possibility of specifying any desired pooling ratio adaptable to the size $N_i$ of the $i$-th graph, *i.e.*, $K_i = \lfloor N_i * 0.5 \rfloor$, while achieving node selection patterns similar to one-every-$K$ methods. In methods like $k$-MIS, the flexibility provided by trainable functions is only used to choose between a small set of maximal solutions that do not break the hard constraint of the supernodes to be $K$-independent. On the other hand, in MaxCutPool any set of supernodes can be chosen in principle, making MaxCutPool more flexible and able to adapt to the requirements of the downstream task. Finally, MaxCutPool is the only scoring-based pooler with a graph-theoretical auxiliary regularization loss.

The proposed method for constructing the cluster assignment matrix $S$ from the selected supernodes can be applied to other scoring-based pooling methods. This naturally enhances their expressiveness by enabling the use of the same CON and RED operations adopted by soft-clustering approaches, which retain all the information from the original graph. By addressing the key limitation in the expressiveness of scoring-based pooling methods, our approach retains the benefits of sparsity and interpretability of scoring-based poolers while narrowing the gap with soft-clustering methods.

## 4 EXPERIMENTAL EVALUATION

We consider three different tasks to demonstrate the effectiveness of MaxCutPool. The code to reproduce the reported results is publicly available[1]. The details of the architectures used in each experiment and the hyperparameter selection procedure are described in Appendix C.

### 4.1 COMPUTATION OF THE MAXCUT PARTITION

The main focus of this experiment is to evaluate the capability of the proposed loss $\mathcal{L}_{\text{cut}}$ to optimize the MAXCUT objective, despite the potential risks of getting stuck in local minima due to its gradient-based nature. We compute a MAXCUT partition with a simple GNN consisting of a MP layer followed by MaxCutPool, which is trained by minimizing only the loss in Eq. 5 (details in Appendix C.2). We compare our model against the LEVS approach based on $\boldsymbol{u}_{\max}$, the GW algorithm, and a GNN with GCN layers that minimize a MAXCUT loss, as proposed by Schuetz et al. (2022). For a fair comparison, ours and the latter GNN architecture have a comparable number of learnable parameters.

We considered 9 graphs generated via the PyGSP library (Defferrard et al., 2017), including bipartite graphs such as the Grid2D and Ring, and 7 graphs from the GSet dataset (Ye, 2003), including random, planar, and toroidal graphs, typically used as benchmarks for evaluating MAXCUT algorithms (details in App. D.1). Results are shown in Tab. 2. Performance is computed in terms of the percentage of cut edges: the higher, the better. With one exception, MaxCutPool always finds the best cut.

---

[1] https://github.com/NGMLGroup/MaxCutPool

Table 2: Size of the graph cuts obtained with MaxCutPool, a GNN with GCN layers, and two common algorithms to compute the MAXCUT. GW results are absent for some entries of the PyGSP datasets and for GSet because the solver failed to converge.

(a) PyGSP datasets

| Dataset | GW | NDP | GCN | MaxCutPool |
|---|---|---|---|---|
| BarabasiAlbert | 0.6875 | 0.6589 | 0.7240 | **0.7292** |
| Community | 0.6767 | 0.6429 | 0.6805 | **0.6814** |
| ErdősRenyi | 0.6920 | 0.6858 | 0.6797 | **0.7105** |
| Grid (10×10) | **1.0000** | **1.0000** | 0.9222 | **1.0000** |
| Grid (60×40) | - | 0.9787 | 0.1862 | **0.9815** |
| Minnesota | - | 0.9104 | 0.8904 | **0.9130** |
| RandRegular | 0.4827 | 0.8760 | 0.8733 | **0.9040** |
| Ring | **1.0000** | **1.0000** | 0.4200 | **1.0000** |
| Sensor | 0.6000 | 0.5719 | 0.6281 | **0.6406** |

(b) GSet datasets

| Dataset | NDP | GCN | MaxCutPool |
|---|---|---|---|
| G14 | 0.6155 | 0.6323 | **0.6412** |
| G15 | 0.5945 | 0.6288 | **0.6424** |
| G22 | 0.6441 | 0.6409 | **0.6577** |
| G49 | **1.0000** | 0.9683 | **1.0000** |
| G50 | **0.9800** | 0.9610 | 0.9750 |
| G55 | 0.7568 | 0.7865 | **0.8068** |
| G70 | 0.8803 | 0.8945 | **0.9086** |

## 4.2 GRAPH CLASSIFICATION

For this task, we evaluate the classification accuracy of a GNN classifier with the following structure: MP(32)-Pool-MP(32)-Readout, using Graph Isomorphism Network (GIN) (Xu et al., 2019) as the MP layer. The Pool operation is implemented either by MaxCutPool or by the following competing methods: Diffpool (Ying et al., 2018), DMoN (Tsitsulin et al., 2023), MinCutPool (Bianchi et al., 2020a), Top-$k$ (Gao & Ji, 2019), Graclus (Dhillon et al., 2007), $k$-MIS (Bacciu et al., 2023). We also consider Edge-Contraction Pooling (ECPool) (Diehl, 2019) that pools the graph by contracting the edge connecting similar nodes. For MaxCutPool, we evaluate three variants: (i) MaxCutPool, the standard version; (ii) MaxCutPool-E, the variant with expressive CON; (iii) MaxCutPool-NL, where "NL" stands for "no loss", meaning we do not optimize the auxiliary loss in the GNN. This serves as an ablation study to assess the importance of the auxiliary loss. Whenever edge attributes are available, the first GIN layer is replaced by a GINE layer (Hu et al., 2020), which takes into account edge attributes. Further implementation details are in Appendix C.3.

As graph classification datasets we consider 8 TUD datasets (COLLAB, DD, NCI1, ENZYMES, MUTAG, Mutagenicity, PROTEINS, and REDDIT-BINARY) (Morris et al., 2020), the Graph Classification Benchmark Hard (GCB-H) (Bianchi et al., 2022), and EXPWL1 (Bianchi & Lachi, 2023), which is a recent dataset for testing the expressive power of GNNs. In addition, we introduce a novel dataset consisting of 5,000 multipartite graphs: each graph is complete 10-partite, meaning that the nodes can be partitioned into 10 groups so that the nodes in each group are disconnected, but are connected to all the nodes of the other groups. To the best of our knowledge, this is the first benchmark dataset for graph classification with heterophilic graphs. While the Multipartite dataset consists of complex graph structures, the classification label is determined solely by the node features, allowing us to assess whether the GNN can effectively isolate relevant information despite the presence of misleading topological information. The construction of the Multipartite dataset and a further discussion about its properties are reported in Appendix D.2.

Whenever the node features were not available, we used node labels. If node labels were also unavailable, we used a constant as a surrogate node feature. Further details about the remaining datasets can be found in Appendix D.3. The datasets were split via a 10-fold cross-validation procedure. The training dataset was further partitioned into a 90-10% train-validation random split. This approach is similar to the procedure described by Errica et al. (2020). Each model was trained for 1,000 epochs with early stopping, keeping the checkpoint with the best validation accuracy.

The results are reported in Tab. 3. For completeness, we also reported the performance of the same GNN model without pooling layers ("No pool"). We conducted a preliminary ANOVA test ($p$-value 0.05) for each dataset followed by a pairwise Tukey-HSD test ($p$-value 0.05) to group models whose performance is not significantly different. Those belonging to the top-performing group are colored green. The ANOVA test failed on ENZYMES, PROTEINS, MUTAG, and DD, meaning that the

Table 3: Mean and standard deviations of the graph classification accuracy. For each dataset the best performing method and those that are not significantly different from it are colored in green. If a method is in the top-performing group is assigned with a score of 1, 0 otherwise.

| Pooler | GCB-H | COLLAB | EXPWL1 | Mult. | Mutag. | NCI1 | REDDIT-B | Score |
|---|---|---|---|---|---|---|---|---|
| No pool | $74_{\pm4}$ | $74_{\pm2}$ | $87_{\pm2}$ | $14_{\pm12}$ | $79_{\pm2}$ | $78_{\pm3}$ | $90_{\pm2}$ | - |
| DiffPool | $51_{\pm8}$ | $70_{\pm2}$ | $69_{\pm3}$ | $9_{\pm1}$ | $78_{\pm2}$ | $75_{\pm2}$ | $90_{\pm2}$ | 1 |
| DMoN | $74_{\pm3}$ | $68_{\pm2}$ | $73_{\pm3}$ | $52_{\pm2}$ | $80_{\pm2}$ | $77_{\pm2}$ | $88_{\pm2}$ | 3 |
| EdgePool | $75_{\pm4}$ | $72_{\pm3}$ | $90_{\pm2}$ | $55_{\pm3}$ | $80_{\pm2}$ | $77_{\pm3}$ | $91_{\pm2}$ | 4 |
| Graclus | $75_{\pm3}$ | $72_{\pm3}$ | $90_{\pm2}$ | $25_{\pm18}$ | $80_{\pm2}$ | $77_{\pm2}$ | $90_{\pm3}$ | 4 |
| $k$-MIS | $75_{\pm4}$ | $71_{\pm2}$ | $99_{\pm1}$ | $58_{\pm2}$ | $79_{\pm2}$ | $75_{\pm3}$ | $90_{\pm2}$ | 4 |
| MinCutPool | $75_{\pm5}$ | $70_{\pm2}$ | $71_{\pm3}$ | $56_{\pm3}$ | $78_{\pm3}$ | $73_{\pm3}$ | $87_{\pm2}$ | 1 |
| Top-$k$ | $56_{\pm5}$ | $72_{\pm2}$ | $73_{\pm2}$ | $43_{\pm3}$ | $75_{\pm3}$ | $73_{\pm2}$ | $77_{\pm2}$ | 0 |
| MaxCutPool | $73_{\pm3}$ | $77_{\pm2}$ | $100_{\pm0}$ | $90_{\pm2}$ | $77_{\pm2}$ | $75_{\pm2}$ | $89_{\pm3}$ | 5 |
| MaxCutPool-E | $74_{\pm3}$ | $77_{\pm2}$ | $100_{\pm0}$ | $87_{\pm5}$ | $79_{\pm1}$ | $76_{\pm2}$ | $89_{\pm2}$ | 7 |
| MaxCutPool-NL | $61_{\pm6}$ | $77_{\pm3}$ | $100_{\pm0}$ | $91_{\pm1}$ | $76_{\pm3}$ | $74_{\pm2}$ | $86_{\pm3}$ | 3 |

difference in the performance of the GNNs equipped with different poolers is not significant. For this reason, the results on these datasets are omitted from Tab. 3 and reported in Appendix E.1.

MaxCutPool consistently ranks among the top-performing methods across all evaluated datasets. Notably, on the EXPWL1 even the non-expressive variant of MaxCutPool achieves a perfect accuracy (100%), outperforming the competitors. This is the first known example of a non-expressive pooler passing the expressiveness test provided by this dataset. On the Multipartite dataset, MaxCutPool performs significantly better than every pooling method. When compared to the "No pool" baseline, on most datasets MaxCutPool improves the classification performance by increasing the receptive field of the MP layers while retaining only the necessary information and enhancing the overall expressive power of the GNN model. It is worth noting that the EXPWL1 and Multipartite are the least homophilic datasets (see Appendix D.3), indicating that MaxCutPool is particularly effective for heterophilic graphs. On the COLLAB dataset, all MaxCutPool variants achieve the top accuracy of 77%, showing a statistically significant improvement over other methods. Notably, in the MaxCutPool and MaxCutPool-E variants the auxiliary loss term plateaued around 0, making them equivalent to the MaxCutPool-NL variant that, in this case, achieves the same performance. This indicates that our method remains robust even when the auxiliary loss is not needed for the downstream task. Overall, the MaxCutPool-E variant, which satisfies expressiveness conditions, exhibits similar or better performance compared to MaxCutPool across most datasets. In contrast, the performance decline observed in the MaxCutPool-NL variant demonstrates the importance of the auxiliary loss.

## 4.3 NODE CLASSIFICATION

For this task, we adopted a simple auto-encoder architecture for node classification: MP(32)-Pool-MP(32)-Unpool-MP(32)-Readout, with GIN as MP. The Unpool operation (also referred to as *lifting* (Jin et al., 2020)) is implemented by copying into each node $i$ the value of the supernode $j$ to which node $i$ was assigned by the SEL operation in the pooling phase. Zero-padding is used when lifting nodes not assigned to any supernode, like in the case of Top-$k$. Further details about the architecture for node classification and the unpooling procedure are deferred to Appendix C.4.

For this experiment, we considered the 5 heterophilic datasets presented in Platonov et al. (2023) (details in Appendix D.4). As pooling methods we considered Top-$k$ (Gao & Ji, 2019), $k$-MIS (Bacciu et al., 2023), NDP (Bianchi et al., 2020b), and MaxCutPool. We did not consider Graclus or any soft-clustering poolers, as they were exhausting the RAM and GPU VRAM, respectively, given the large size of the graphs. On the other hand, MaxCutPool is very parsimonious in terms of computational resources and scales very well with the graph size. To systematically estimate the space complexity of the different pooling methods, we performed an experimental evaluation of the GPU VRAM usage, which can be found in Appendix F.3.

Table 4: Node classification accuracy (Roman-empire, Amazon-ratings) and AUROC (Minesweeper, Tolokers, Questions). The best performing models in each dataset are in green and get 1 score point, 0 otherwise.

| Pooler | Roman-e. | Amazon-r. | Minesw. | Tolokers | Questions | Score |
|---|---|---|---|---|---|---|
| No pool | $59_{\pm 0}$ | $46_{\pm 1}$ | $86_{\pm 2}$ | $86_{\pm 4}$ | $71_{\pm 2}$ | - |
| Top-$k$ | $26_{\pm 7}$ | $46_{\pm 4}$ | $94_{\pm 1}$ | $89_{\pm 5}$ | $64_{\pm 3}$ | 1 |
| $k$-MIS | $23_{\pm 3}$ | $48_{\pm 2}$ | $75_{\pm 2}$ | $84_{\pm 2}$ | $83_{\pm 1}$ | 1 |
| NDP | $22_{\pm 5}$ | $53_{\pm 2}$ | $98_{\pm 0}$ | $88_{\pm 6}$ | $68_{\pm 4}$ | 3 |
| MaxCutPool | $56_{\pm 3}$ | $53_{\pm 1}$ | $96_{\pm 1}$ | $87_{\pm 3}$ | $82_{\pm 4}$ | 4 |
| MaxCutPool-E | $60_{\pm 4}$ | $53_{\pm 2}$ | $97_{\pm 1}$ | $91_{\pm 2}$ | $85_{\pm 5}$ | 5 |

Following Platonov et al. (2023), in Tab. 4 we report the means and standard deviations of the accuracy for Roman-empire and Amazon-ratings, and of the ROC AUC for Tolokers, Minesweeper, and Questions. The results are computed on the 10 public folds of these datasets. When configured with MaxCutPool and MaxCutPool-E, the node classification architecture achieves significantly superior performance on the Roman-Empire dataset, which is notably the most heterophilic among all the datasets (see Tab. 11). Also on the remaining datasets, our method performs well: unlike the other pooling methods that achieve top performance only on a subset of the datasets, MaxCutPool-E is consistently in the top tier.

## 5 CONCLUSION

This work contributes significantly to both the MAXCUT optimization and the development of specialized GNN architectures to solve combinatorial optimization problems. Our proposed GNN-based MAXCUT algorithm not only extends the MAXCUT optimization to attributed graphs and combines it with task-specific losses but also surpasses the performances of traditional methods on non-attributed graphs. While a conventional GNN with a huge capacity manages to optimize a MAXCUT loss (Schuetz et al., 2022), our model is much more efficient thanks to the Heterophilic Message Passing layers. These results highlight the importance of aligning the GNN architecture with the problem's inherent structure: in this case, leveraging heterophilic propagation to solve problems that seek dissimilarity between neighboring nodes.

Our second contribution is to utilize the proposed MAXCUT optimizer to implement a graph pooling method that combines the flexibility of soft-clustering approaches with the efficiency of scoring-based methods and with the theoretically-inspired design of one-every-$K$ strategies. GNNs for graph and node classification equipped with our proposed pooling layer consistently achieves superior performance across diverse downstream tasks. Unlike existing graph pooling and graph coarsening approaches that aim at preserving low-frequencies on the graph (Loukas, 2019), our method performs exceptionally well also on heterophilic datasets.

While our pooling layer can implement any pooling ratio, the auxiliary loss is optimized for the node partition induced by the MAXCUT, whose size might not be aligned with the specified pooling ratio. When the distribution of the nodes' degree is approximately uniform, the MAXCUT induces an approximately balanced partition corresponding to a pooling ratio of $\approx 0.5$, which is, thus, generally a good choice.

Looking forward, we see great potential in pretraining GNNs with auxiliary losses. This aligns with the principles of foundational models (Bommasani et al., 2021) and could facilitate the development of more effective and general-purpose graph pooling techniques.

ACKNOWLEDGMENTS

This work was supported by the Norwegian Research Council project 345017: *RELAY: Relational Deep Learning for Energy Analytics*. The authors wish to thank Nvidia Corporation for donating some of the GPUs used in this project.

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

# APPENDIX

## A    NEAREST NEIGHBOR ASSOCIATION ALGORITHM

As discussed in Sec. 3, each node is associated with one of the supernodes (preferably the closest in terms of path-distance on the graph). Naively searching for each node the closest supernode is computationally demanding and becomes intractable for large graphs. Therefore, we propose an implementation of the assignment scheme that is efficient and can be easily parallelized on a GPU. The proposed algorithm is based on a Breadth First Search (BFS) of the graph and is detailed in the pseudo-code in Algorithm 1.

---

**Algorithm 1** Pseudo-code for the assignment scheme to the supernodes

---

1:  **procedure** ASSIGNNODESTOSUPERNODES($\mathcal{G}, \mathcal{S}, MaxIter$)
2:      $\boldsymbol{E} \leftarrow$ InitializeEncodings($\mathcal{G}, \mathcal{S}$)                          ▷ One-hot encoding
3:      $\boldsymbol{m} \leftarrow$ InitializeMask($\mathcal{G}, \mathcal{S}$)
4:      $Assignments \leftarrow$ InitializeEmptyList()
5:      **for** $i = 1$ to $MaxIter$ **do**
6:          **if** AllNodesAssigned($\boldsymbol{m}$) **then**
7:              **break**
8:          **end if**
9:          $\boldsymbol{E}' \leftarrow$ ParallelMessagePassing($\mathcal{G}, \boldsymbol{E}$)                          ▷ $\boldsymbol{E}' = \boldsymbol{A}\boldsymbol{E}$
10:         $Assignments \leftarrow$ ParallelAssignment($\boldsymbol{E}', \mathcal{S}, \boldsymbol{m}$)
11:         $\boldsymbol{m} \leftarrow$ UpdateMask($\boldsymbol{m}, Assignments$)
12:         $\boldsymbol{E} \leftarrow \boldsymbol{E}'$
13:     **end for**
14:     **if** not AllNodesAssigned($\boldsymbol{m}$) **then**
15:         $RndAssignments \leftarrow$ ParallelRandomAssignment($UnassignedNodes, \mathcal{S}$)
16:     **end if**
17:     $FinalAssignments \leftarrow$ GetFinalAssignments($Assignments, RndAssignments$)
18:     **return** $FinalAssignments$
19: **end procedure**

---

The algorithm takes as input the graph $\mathcal{G}$ (in particular, its topology described by the adjacency matrix $\boldsymbol{A}$), the set of $K$ supernodes $\mathcal{S}$ identified by the `SEL` operation, and a maximum number of iterations ($MaxIter$), which represent the maximum number of steps a node can traverse the graph to reach its closest supernode before being assigned at random.

In line 2, an encoding matrix $\boldsymbol{E}$ of size $N \times K + 1$ is initialized so that row $i$ is a one-hot vector with the non-zero entry in position $k + 1$, if the node $i$ of the original graph is the $k$-th supernode. Otherwise, row $i$ in a zero-vector of size $K + 1$. This matrix will be gradually populated when supernodes are encountered during the BFS. It's important to note that the 0-th column in matrix $\boldsymbol{E}$ (and subsequently in $\boldsymbol{E}'$) serves a special purpose. This column represents a "fake" supernode, which plays a crucial role in the assignment process.

A Boolean mask $\boldsymbol{m} \in \{0, 1\}^N$ indicating whether a node has already encountered the closest supernode is initialized in line 3 with 1 in position $i$ if node $i$ is a supernode and 0 otherwise. Finally, an empty list indicating to which supernode each node is assigned is initialized (line 4).

Until the maximum number of iterations is reached or until all nodes are assigned (line 6), the encoding matrix $\boldsymbol{E}$ is propagated with an efficient message passing operation (line 9) that can be parallelized on a GPU. As soon as a 1 appears in position $k$ within a line $i$ of $\boldsymbol{E}$ previously full of zeros, node $i$ is assigned to supernode $k$ and the assignments and mask $\boldsymbol{m}$ are updated accordingly (lines 10 and 11). The *ParallelAssignment* function (line 10), in particular, takes the rows of the newly generated embeddings $\boldsymbol{E}'$ that have not yet been assigned and performs an `argmax` operation on the last dimension. If the `argmax` doesn't find any valid supernode for a node (*i.e.*, all values in the row are zero), it returns 0, effectively assigning the node to the "fake" supernode represented by the 0-th column. This allows to filter out the unassigned nodes in line 11.

If there are still unassigned nodes at the end of the iterations, the remaining nodes are randomly assigned to one of the $K$ supernodes (line 15). Finally, all the assignments are merged (line 17).

## B   DERIVATION OF THE AUXILIARY LOSS

Let us consider the MAXCUT objective in Equation 1. It can be rewritten as

$$\max_{\boldsymbol{z}} \left( \sum_{i,j \in \mathcal{V}} w_{ij} - \sum_{i,j \in \mathcal{V}} z_i z_j w_{ij} \right) = \max_{\boldsymbol{z}} \left( |\mathcal{E}| - \sum_{i,j \in \mathcal{V}} z_i z_j w_{ij} \right),$$

which is equivalent to

$$\max_{\boldsymbol{z}} \left( 1 - \sum_{i,j \in \mathcal{V}} \frac{z_i z_j w_{ij}}{|\mathcal{E}|} \right).$$

The solution $\boldsymbol{z}^*$ for the original objective is thus the solution for

$$\min_{\boldsymbol{z}} \frac{\boldsymbol{z}^\top A \boldsymbol{z}}{|\mathcal{E}|}.$$

## C   IMPLEMENTATION DETAILS

### C.1   MAXCUTPOOL LAYER AND SCORENET

A schematic depiction of the MaxCutPool layer is illustrated in Fig. 3, where the SEL, RED, and CON operations are highlighted.

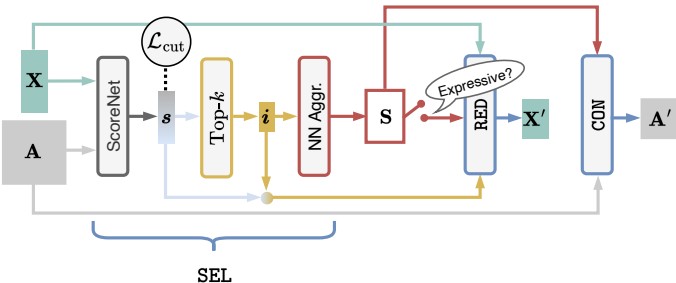

Figure 3: Scheme of the MaxCutPool layer.

SEL first computes a score vector $\boldsymbol{s}$ using the auxiliary GNN, ScoreNet, based on the node features $\mathbf{X}$ and the connectivity matrix $\mathbf{A}$. A top-$K$ operation is used to find the indices $\boldsymbol{i}$ of the $K$ nodes with the highest scores that become the supernodes, *i.e.*, the nodes of the pooled graph. The remaining $N - K$ nodes are assigned to the nearest supernode through a nearest-neighbor (NN) aggregation procedure that yields an assignment matrix $\mathbf{S}$, whose $jk$-th element is 1 if node $j$ is assigned to supernode $k$, and zero otherwise. The score vector $\boldsymbol{s}$ is used to compute the loss $\mathcal{L}_{\text{cut}}$, which is associated with each MaxCutPool layer.

The RED operation computes the node features of the pooled graph $\boldsymbol{X}'$ by multiplying the features of the selected nodes $\boldsymbol{X}_i$ with the scores $\boldsymbol{s}$. This operation is necessary to let the gradients flow past the top-$K$ operation, which is not differentiable. In the expressive variant, MaxCutPool-E, RED computes the new node features by combining those from all the nodes in the graph through the multiplication with matrix $\boldsymbol{S}$. We combine the features by summing them instead of taking the average since the sum enhances the expressiveness of the pooling layer (Bianchi & Lachi, 2023).

The CON operations always leverage the assignment matrix to compute the adjacency matrix of the pooled graph. In particular, the edge connecting two supernodes $i$ and $j$ is obtained by coalescing all the edges connecting the nodes assigned to supernode $i$ with those assigned to supernode $j$. Also in this case, we take the sum as the operation to coalesce the edges. The resulting edges in the pooled graph are associated with a weight $w_{ij}$ that counts the number of combined edges.

The details of the ScoreNet used in the MaxCutPool layer are depicted in Fig. 4. The ScoreNet consists of a linear layer that maps the features $X$ to a desired hidden dimension. Afterward, a stack of HetMP layers gradually transforms the node features by amplifying their high-frequency components with heterogeneous MP operations. Finally, an MLP transforms the node features of the last HetMP layer into a score vector $s$, which is a high-frequency graph signal. We note that while

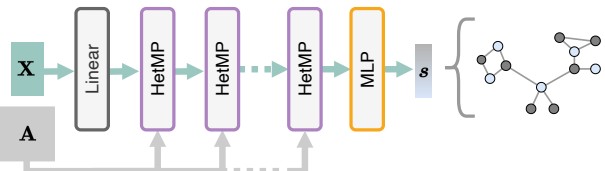

Figure 4: Scheme of the ScoreNet.

the simple HetMP we adopted works well in our case, different heterophilic MP operators could have been considered (Chien et al., 2021; Dong et al., 2021; Fu et al., 2022).

The ScoreNet is configured with the following hyperparameters:

- Number of HetMP layers and number of features in each layer. We use the notation $[32, 16, 8]$ to indicate a ScoreNet with three HetMP layers with hidden sizes 32, 16, and 8, respectively. We also use the notation $[32] \times 4$ to indicate 4 layers with 32 units each. As default, we use $[32, 32, 32, 32, 16, 16, 16, 16, 8, 8, 8, 8]$.
- Activation function of the HetMP layers. As the default, we use TanH.
- Number of layers and features in the MLP. As default, we use $[16, 16]$.
- Activation function of the MLP. As the default, we use ReLU.
- Smoothness hyperparameter $\delta$. As default, we use 2.
- Auxiliary loss weight $\beta$. As default, we use 1.

The optimal configuration has been identified with the cross-validation procedure described in Sec. 4. Depending on the experiment and the GNN architecture, some parameters in the ScoreNet are kept fixed at their default value while others are optimized.

## C.2 CUT MODEL

The model used to compute the MAXCUT is depicted in Fig. 5. The model consists of a single MP layer followed by the ScoreNet, which returns the score vector $s$. The MAXCUT partition is obtained

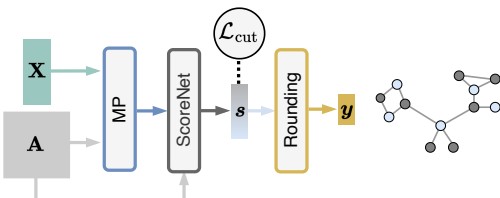

Figure 5: Scheme of the model used for computing the MAXCUT.

by rounding the values in the score vector as follows

$$y_i = \begin{cases} 1 & \text{if } s_i > 0, \\ -1 & \text{otherwise.} \end{cases}$$

The model is trained in a completely unsupervised fashion only by minimizing the auxiliary loss $\mathcal{L}_{\text{cut}}$.

As MP layers, we used a GIN (Xu et al., 2019) layer with 32 units and ELU activation function. The model was trained for 2000 epochs with Adam optimizer (Kingma & Ba, 2015), with the initial learning rate set to $8e-4$. We used a learning rate scheduler that reduces by $0.8$ the learning rate when the auxiliary loss does not improve for 100 epochs. For testing, we restored the model checkpoint that achieved the lowest auxiliary loss.

The best configuration was found via a grid search on the following set of hyperparameters:

- HetMP layers and units:
  - $[32] \times 4$,
  - $[4] \times 32$,
  - $[8] \times 16$,
  - $[16] \times 8$,
  - $[32, 32, 32, 32, 16, 16, 16, 16, 8, 8, 8, 8]$.
- HetMP activations:
  - ReLU,
  - TanH.
- Smoothness hyperparameter $\delta$:
  - 2,
  - 3,
  - 5.

In Tab. 5 we report the configurations of the ScoreNet used for the different graphs in the MAXCUT experiment.

Table 5: Hyperparameters configurations of the ScoreNet for the MAXCUT task.

| Dataset | MP units | MP Act | $\delta$ |
|---|---|---|---|
| G14 | $[32, 32, 32, 32, 16, 16, 16, 16, 8, 8, 8, 8]$ | ReLU | 2.0 |
| G15 | $[32, 32, 32, 32, 16, 16, 16, 16, 8, 8, 8, 8]$ | ReLU | 2.0 |
| G22 | $[4] \times 32$ | TanH | 2.0 |
| G49 | $[32, 32, 32, 32, 16, 16, 16, 16, 8, 8, 8, 8]$ | TanH | 2.0 |
| G50 | $[8] \times 16$ | ReLU | 2.0 |
| G55 | $[4] \times 32$ | ReLU | 2.0 |
| G70 | $[8] \times 16$ | ReLU | 2.0 |
| BarabasiAlbert | $[4] \times 32$ | TanH | 2.0 |
| Community | $[4] \times 32$ | TanH | 2.0 |
| ErdősRenyi | $[4] \times 32$ | TanH | 2.0 |
| Grid2d ($10\times10$) | $[4] \times 32$ | TanH | 2.0 |
| Grid2d ($60\times40$) | $[4] \times 32$ | ReLU | 2.0 |
| Minnesota | $[4] \times 32$ | TanH | 2.0 |
| RandRegular | $[4] \times 32$ | TanH | 2.0 |
| Ring | $[4] \times 32$ | ReLU | 2.0 |
| Sensor | $[4] \times 32$ | TanH | 2.0 |

## C.3 GRAPH CLASSIFICATION MODEL

The model used to perform graph classification is depicted in Fig. 6. The model consists of an MP layer, followed by a pooling layer, an MP acting on the pooled graph, a global pooling layer that sums

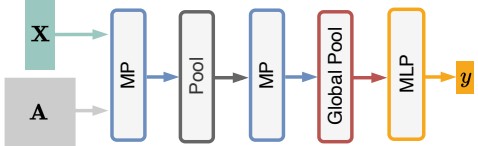

Figure 6: Scheme of the graph classification model.

the features of all the nodes in the pooled graph, and an MLP that produces the label $y$ associated with the input graph. The model is trained by jointly minimizing the cross-entropy loss between the predicted graph labels and the true ones and the auxiliary losses associated with the different pooling layers. The models are trained using a batch size of 32 for 1000 epochs, using the Adam optimizer with an initial learning rate of $1e - 4$. We used an early stopping that monitors the validation loss with a patience of 300 epochs. For testing, we restored the model checkpoint that achieved the lowest validation loss during training.

The best configuration was found via a grid search on the following set of hyperparameters:

- HetMP layers and units:
  - $[32] \times 8$,
  - $[32] \times 4$,
  - $[8] \times 16, [16] \times 8$,
  - $[32, 32, 16, 16, 8, 8]$,
  - $[32, 32, 32, 32, 16, 16, 16, 16, 8, 8, 8, 8]$.
- Auxiliary loss weight $\beta$:
  - 1,
  - 2,
  - 5.

Table 6: Hyperparameters configurations of the ScoreNet for the graph classification task.

| | **MaxCutPool** | | **MaxCutPool-E** | |
|---|---|---|---|---|
| **Dataset** | **MP units** | $\beta$ | **MP units** | $\beta$ |
| GCB-H | $[8] \times 16$ | 3.0 | $[32] \times 8$ | 5.0 |
| COLLAB | $[32] \times 8$ | 1.0 | $[32] \times 8$ | 1.0 |
| DD | $[32, 32, 32, 32, 16, 16, 16, 16, 8, 8, 8, 8]$ | 1.0 | $[8] \times 16$ | 5.0 |
| ENZYMES | $[8] \times 16$ | 3.0 | $[16] \times 8$ | 3.0 |
| EXPWL1 | $[32, 32, 16, 16, 8, 8]$ | 1.0 | $[16] \times 8$ | 1.0 |
| MUTAG | $[8] \times 16$ | 1.0 | $[16] \times 8$ | 3.0 |
| Multipartite | $[32] \times 8$ | 3.0 | $[32] \times 8$ | 1.0 |
| Mutagenicity | $[32, 32, 16, 16, 8, 8]$ | 1.0 | $[32] \times 8$ | 5.0 |
| NCI1 | $[32, 32, 16, 16, 8, 8]$ | 1.0 | $[8] \times 16$ | 3.0 |
| PROTEINS | $[32, 32, 32, 32, 16, 16, 16, 16, 8, 8, 8, 8]$ | 3.0 | $[32, 32, 16, 16, 8, 8]$ | 5.0 |
| REDDIT-B | $[32] \times 8$ | 1.0 | $[32, 32, 32, 32, 16, 16, 16, 16, 8, 8, 8, 8]$ | 1.0 |

In Tab. 6 we report the configurations of the ScoreNet used in the graph classification architecture for the different datasets in the expressive and non-expressive variant of MaxCutPool.

## C.4 Node classification model

The model used to perform node classification is depicted in Fig. 7. The model consists of an MP layer, followed by a pooling layer, an MP acting on the pooled graph, an unpooling (lifting) layer, an MP on the unpooled graph, and an MLP that produces the final node labels $y$.

The entry $y_i$ represents the predicted label for node $i$. The model is trained by jointly minimizing the cross-entropy loss between the predicted node labels and the true ones and the auxiliary loss of the pooling layer.

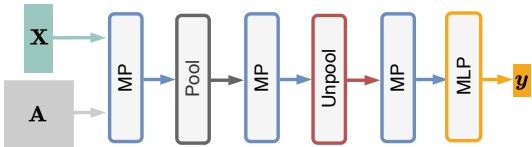

Figure 7: Scheme of the node classification model.

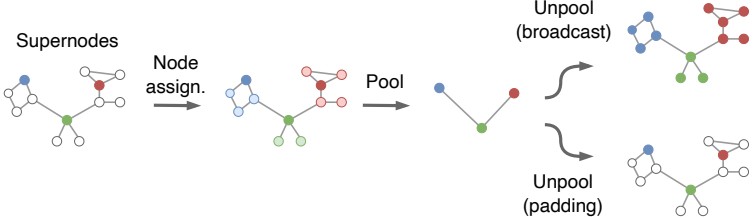

Figure 8: The two possible strategies for performing unpooling (lifting).

When using MaxCutPool, the unpooling/lifting procedure can be implemented in two different ways, illustrated in Fig. 8. The first strategy, *broadcast* unpooling, copies the values $\boldsymbol{X}'$ of the nodes of the pooled graph to both the corresponding supernodes and to the nodes associated with the supernodes according to the assignment matrix $\boldsymbol{S}$, obtained as described in Sec. 3 and Appendix A. Formally, the unpooled node features $\tilde{\boldsymbol{X}}$ are:

$$\tilde{\boldsymbol{X}} = \boldsymbol{S}\boldsymbol{X}'.$$

We note that this is the commonly used approach to perform unpooling in cluster-based poolers.

In the second strategy, *padding* unpooling, the values $\boldsymbol{X}'$ are copied back only to the supernodes, while the remaining nodes are padded with a zero-valued vector:

$$[\tilde{\boldsymbol{X}}]_i = \begin{cases} [\boldsymbol{X}']_i & \text{if } i \text{ is a supernode} \\ \boldsymbol{0} & \text{otherwise.} \end{cases}$$

This is the approach to perform unpooling used by scoring-based approaches such as Top-$k$ and by one-over-$K$ approaches such as NDP that only select the supernodes and leave the remaining nodes unassigned.

For the node classification task presented in Sec. 4.3, as MP layers we used a GIN (Xu et al., 2019) layer with 32 units and ReLU activation function. The MLP has a single hidden layer with 32 units, a ReLU activation function, and a dropout layer between the hidden and output layers with a dropout probability of $0.1$. The unpooling strategy used in this architecture is the broadcast one for $k$-MIS and MaxCutPool and the padding one for Top-$k$ and NDP.

The best configuration was found via a grid search on the following set of hyperparameters:

- HetMP layers and units:
  - $[32] \times 4$,
  - $[4] \times 32$,
  - $[32, 32, 32, 32, 16, 16, 16, 16, 8, 8, 8, 8]$.
- MLP activations:
  - ReLU,
  - TanH.

The configuration of the ScoreNet for the MaxCutPool pooler used in the different datasets is reported in Tab. 7.

The node classifier was trained for $20,000$ epochs, using the Adam optimizer with an initial learning rate of $5e - 4$. We used a learning rate scheduler that reduces by $0.5$ the learning rate when the

Table 7: Hyperparameter configurations of the ScoreNet in the node classification task.

| Dataset | MP units | MLP Act. |
|---|---|---|
| Roman-Empire | $[32, 32, 32, 32]$ | ReLU |
| Amazon-Ratings | $[32, 32, 32, 32]$ | ReLU |
| Minesweeper | $[32, 32, 32, 32, 16, 16, 16, 16, 8, 8, 8, 8]$ | ReLU |
| Tolokers | $[32, 32, 32, 32, 16, 16, 16, 16, 8, 8, 8, 8]$ | ReLU |
| Questions | $[32, 32, 32, 32]$ | ReLU |

validation loss does not improve for $500$ epochs. We used an early stopping that monitors the validation loss with a patience of $2,000$ epochs. For testing, we restored the model checkpoint that achieved the lowest validation loss during training.

We also considered an additional architecture for node classification with skip (residual) connections, depicted in Fig. 9. This architecture is similar to the Graph U-Net proposed by Gao & Ji (2019). The

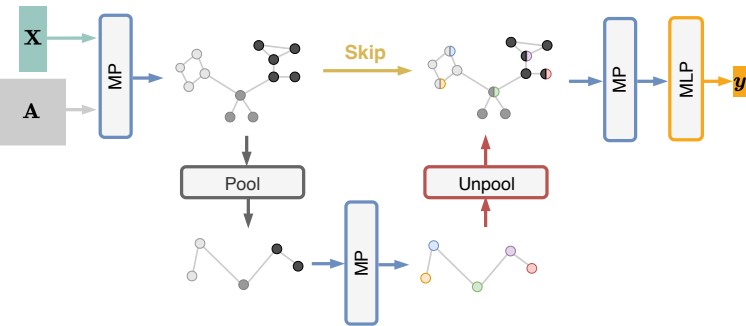

Figure 9: Scheme of the node classification model with skip connections.

node features obtained after the first MP layer are concatenated to the node features generated by the unpooling step. In this architecture, we used the broadcast unpooling for $k$-MIS and padding unpooling for MaxCutPool, Top-$k$, and NDP. The results obtained with this architecture are reported in App. E. In Tab. 8 we report the configurations of the ScoreNet used in the architecture with skip connection in the different datasets.

Table 8: Hyperparameter configurations for the node classification task based on the architecture with skip connections.

| Dataset | MP units | MLP Act. |
|---|---|---|
| Roman-Empire | $[32, 32, 32, 32]$ | ReLU |
| Amazon-Ratings | $[32, 32, 32, 32]$ | ReLU |
| Minesweeper | $[32, 32, 32, 32]$ | TanH |
| Tolokers | $[32, 32, 32, 32]$ | ReLU |
| Questions | $[32, 32, 32, 32]$ | ReLU |

## C.5 IMPLEMENTATION OF OTHER POOLING LAYERS

The pooling methods Top-$k$, Diffpool, DMoN, Graclus, and MinCutPool are taken from PyTorch Geometric (Fey & Lenssen, 2019). For $k$-MIS we used the official implementation [2]. For ECPool, we used the efficient parallel implementation [3] proposed by Landolfi (2022). For NDP we adapted

---

[2] https://github.com/flandolfi/k-mis-pool
[3] https://github.com/flandolfi/edge-pool

to PyTorch the original Tensorflow implementation [4]. All pooling layers were used with the default hyperparameters. Since $k$-MIS does not allow for direct specification of the pooling ratio, we set $k = \lfloor 1/k \rfloor$.

## D  DATASETS DETAILS

### D.1  CUT DATASETS

The statistics of the PyGSP and the Gset datasets used to compute the MAXCUT partition in Sec. 4.1 are reported in Tab. 9. While the PyGSP graphs are built from the library (Defferrard et al., 2017), the Gset dataset is downloaded from the original source [5].

Table 9: Statistics of the PyGSP datasets used to compute the MAXCUT.

(a) PyGSP datasets

| Dataset | # Nodes | # Edges | Vertex attr. |
|---|---|---|---|
| Barabasi-Albert | 100 | 768 | 2 |
| Community | 90 | 532 | 2 |
| Erdős-Renyi | 100 | 974 | 2 |
| Grid2d (10×10) | 100 | 360 | 2 |
| Grid2d (60×40) | 2,400 | 9400 | 2 |
| Minnesota | 2642 | 6608 | 2 |
| RandRegular | 500 | 1500 | 2 |
| Ring | 100 | 200 | 2 |
| Sensor | 64 | 640 | 2 |

(b) Gset

| Dataset | # Nodes | # Edges | Vertex attr. |
|---|---|---|---|
| G14 | 800 | $4,694$ | – |
| G15 | 800 | $4,661$ | – |
| G22 | $2,000$ | $19,990$ | – |
| G49 | $3,000$ | $6,000$ | – |
| G50 | $3,000$ | $6,000$ | – |
| G55 | $5,000$ | $12,468$ | – |
| G70 | $10,000$ | $9,999$ | – |

### D.2  MULTIPARTITE DATASET DESCRIPTION

The Multipartite graph dataset is a synthetic dataset consisting of complete multipartite graphs. The nodes of each graph can be partitioned into $C$ clusters of independent nodes, such that every node is connected to every node belonging to every other cluster. The generation of the graphs and the class labels is formally described by the pseudo-code in Algorithm 2 and is discussed in the following:

1. A set of $C$ cluster centers with 2D coordinates $(x, y)$ is initially arranged in a polygon shape. Each center is associated with a label, *i.e.*, a color.

2. The graph class is determined by the position and the color of the cluster centers. Specifically, the graph class is given by the color of the cluster whose center is on the positive $x$-axis.

3. For each class, we generate multiple graphs using these cluster centers. A graph is created by drawing at random the position of the nodes around each cluster center. The number of nodes per cluster varies randomly up to a maximum. Nodes within a cluster share the same color, which is determined by the cluster center.

4. The topology of each graph is obtained by connecting nodes from one cluster to the nodes of all the other clusters, but not to the nodes of the same cluster. Therefore, a node is connected only to nodes with different colors, making the graphs highly heterophilic.

5. After generating graphs for one class, the cluster centers are rotated, and this rotated configuration is used for the next class. Indeed, each rotation brings a different cluster to the positive $x$-axis.

6. The rotation process continues until the graphs for all the $C$ different classes, whose number is equal to the number of clusters, are generated.

Examples of multipartite graphs obtained for $C = 3$ are shown in Figure 10.

The process depends on a few parameters that determine the number of clusters, the maximum nodes per cluster, and the number of graphs per class, providing control over the dataset size and complexity.

---

[4] https://github.com/danielegrattarola/decimation-pooling
[5] http://web.stanford.edu/~yyye/yyye/Gset/

---

**Algorithm 2** Multipartite graph dataset generation

---

**Input:** num_clusters, max_nodes_per_cluster, graphs_per_class
**Output:** dataset
 1: cluster_centers ← GeneratePolygonVertices(num_clusters)          ▷ Initial arrangement of centers
 2: dataset ← {}
 3: **for** class_label ← 0 to num_clusters - 1 **do**
 4:     **for** 1 to graphs_per_class **do**
 5:         graph ← GenerateMultipartiteGraph(cluster_centers, max_nodes_per_cluster)
 6:         graph.label ← class_label                                      ▷ Label based on current rotation
 7:         Add graph to dataset
 8:     **end for**
 9:     cluster_centers ← RotateClockwise(cluster_centers)               ▷ Rotate for next class
10: **end for**
11: **return** dataset

12: **function** GENERATEMULTIPARTITEGRAPH(cluster_centers, max_nodes_per_cluster)
13:     **for** each center in cluster_centers **do**
14:         num_nodes ← RandomInt(1, max_nodes_per_cluster)
15:         node_positions ← GenerateNodesAroundCenter(center, num_nodes)
16:         node_color ← GetColorForCluster(center)                      ▷ Each cluster has a unique color
17:         AddNodesToGraph(node_positions, node_color)
18:     **end for**
19:     ConnectNodesAcrossClusters()                                     ▷ Create complete multipartite graph
20:     **return** graph
21: **end function**

22: **function** ROTATECLOCKWISE(centers)
23:     **return** [centers[-1]] + centers[:-1]                           ▷ Move last center to front
24: **end function**

---

The specific instance of the dataset used in our experiments has 10 centers, 500 graphs per center, and a maximum of 20 nodes per cluster, and is available online [6].

The Multipartite dataset is intentionally designed so that the class label is determined solely by node features: specifically, the color and one of the 2D coordinates (the node's position along the $x$-axis). Although the graph's topology is structured to ensure that each graph is multipartite, this structure is independent of the class label. This creates an intriguing dichotomy between the graph's topology and its classification labels. In theory, a simple MLP focusing exclusively on node features could accurately solve the classification task, as the graph's topology is essentially irrelevant for determining the correct labels. However, when processed by GNNs, this dataset allows us to explore whether the model can correctly identify and utilize the relevant node features for classification, despite the presence of potentially misleading or noisy topological information. Through this

---

[6] https://zenodo.org/doi/10.5281/zenodo.11616515

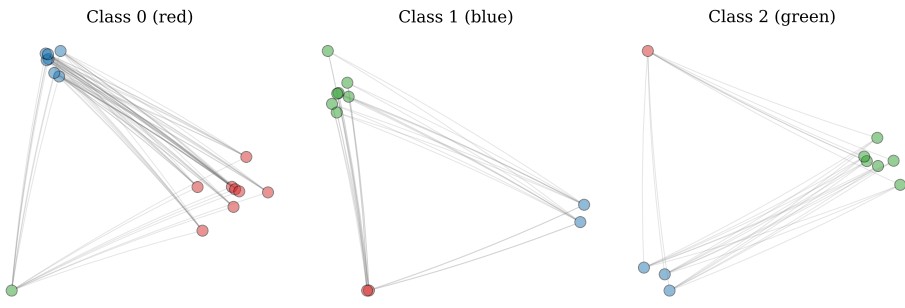

Figure 10: Example of multipartite graphs with $C = 3$ cluster centers generated via our procedure. The graph class corresponds to the color of the nodes from the group to the right.

carefully constructed dataset, we aim to highlight the strengths and potential limitations of certain GNNs architectures, particularly in scenarios where the relationship between graph structure and classification labels is non-trivial, such as in heterophilic datasets.

## D.3 Graph classification datasets

In addition to the novel Multipartite dataset introduced in Sec. D.2, we consider 10 datasets for graph classification in our experimental evaluation. The TU Datasets (Morris et al., 2020) (NCI, PROTEINS, Mutagenicity, COLLAB, REDDIT-B, DD, MUTAG, and Enzymes) are obtained through the loader of PyTorch Geometric [7]. The EXPWL1 and GCB-H datasets, respectively introduced by Bianchi & Lachi (2023) and by Bianchi et al. (2022), are taken from the official repositories [8] [9]. The statistics of each dataset are reported in Tab. 10.

Table 10: Details of the graph classification datasets.

| Dataset | #Samples | #Classes | Avg. #vert. | Avg. #edg. | V. attr. | V. lab. | $\bar{h}(\mathcal{D})$ |
|---|---|---|---|---|---|---|---|
| EXPWL1 | 3,000 | 2 | 76.96 | 186.46 | – | yes | 0.2740 |
| NCI1 | 4,110 | 2 | 29.87 | 64.60 | – | yes | 0.6245 |
| PROTEINS | 1,113 | 2 | 39.06 | 72.82 | 1 | yes | 0.6582 |
| Mutagenicity | 4,337 | 2 | 30.32 | 61.54 | – | yes | 0.3679 |
| COLLAB | 5,000 | 3 | 74.49 | 4,914.43 | – | no | 1 |
| REDDIT-B | 2,000 | 2 | 429.63 | 497.75 | – | no | 1 |
| GCB-H | 1,800 | 3 | 148.32 | 572.32 | – | yes | 0.8440 |
| DD | 1,178 | 2 | 284.32 | 1,431.32 | – | yes | 0.0688 |
| MUTAG | 188 | 2 | 17.93 | 19.79 | – | yes | 0.7082 |
| ENZYMES | 600 | 6 | 32.63 | 62.14 | 18 | yes | 0.6687 |
| Multipartite | 5000 | 10 | 99.79 | 4,477.43 | 3 | yes | 0.1101 |

Since the node labels are not available in the graph classification setting, it is not possible to rely on the homophily ratio $h(\mathcal{G})$ (Lim et al., 2021) considered in the node classification setting. Therefore, to quantify the degree of homophily in the graphs, we look at the node features instead and introduce a surrogate homophily score $\bar{h}(\mathcal{D})$, where $\mathcal{D}$ denotes the whole dataset. The new score is defined as the absolute value of the average cosine similarity between the node features of connected nodes in each graph of the dataset:

$$\bar{h}(\mathcal{D}) = \left| \frac{1}{|\mathcal{D}|} \sum_{\mathcal{G} \in \mathcal{D}} \frac{1}{|\mathcal{E}_{\mathcal{G}}|} \sum_{(i,j) \in \mathcal{E}_{\mathcal{G}}} \frac{\boldsymbol{x}_i \boldsymbol{x}_j}{\|\boldsymbol{x}_i\| \|\boldsymbol{x}_j\|} \right|$$

where $|\mathcal{D}|$ is the number of graphs in the dataset, $\mathcal{E}_{\mathcal{G}}$ is the set of edges of the graph $\mathcal{G}$ and $\boldsymbol{x}_i, \boldsymbol{x}_j$ are the feature vectors of the $i$-th and $j$-th node respectively.

## D.4 Node classification datasets

The datasets are the heterophilic graphs introduced by Platonov et al. (2023) and are loaded with the API provided by PyTorch Geometric [10]. The nodes of each graph are already split in train, validation, and test across 10 different folds. The statistics of the five datasets are reported in Tab. 11. The column $h(\mathcal{G})$ is the class-insensitive edge homophily ratio as defined by Lim et al. (2021), which represents a measure for the level of homophily in the graph.

---

[7] https://pytorch-geometric.readthedocs.io/en/latest/generated/torch_geometric.datasets.TUDataset.html
[8] https://github.com/FilippoMB/The-expressive-power-of-pooling-in-GNNs
[9] https://github.com/FilippoMB/Benchmark_dataset_for_graph_classification
[10] https://pytorch-geometric.readthedocs.io/en/latest/generated/torch_geometric.datasets.HeterophilousGraphDataset.html

Table 11: Statistics of node classification datasets.

| Dataset | # Nodes | # Edges | # Classes | $h(\mathcal{G})$ |
|---|---|---|---|---|
| Roman-Empire | 22,662 | 32,927 | 18 | 0.021 |
| Amazon-Ratings | 24,492 | 93,050 | 5 | 0.127 |
| Minesweeper | 10,000 | 39,402 | 2 | 0.009 |
| Tolokers | 11,758 | 519,000 | 2 | 0.180 |
| Questions | 48,921 | 153,540 | 2 | 0.079 |

# E  ADDITIONAL RESULTS

## E.1  GRAPH CLASSIFICATION

In Tab. 12 we report the additional graph classification results for the dataset where GNNs equipped with different pooling operators did not achieve a significantly different performance from each other.

Table 12: Graph classification accuracy values (subset)

| Pooler | DD | MUTAG | ENZYMES | PROTEINS |
|---|---|---|---|---|
| No pool | $73_{\pm 5}$ | $78_{\pm 13}$ | $33_{\pm 6}$ | $71_{\pm 4}$ |
| Diffpool | $77_{\pm 4}$ | $81_{\pm 11}$ | $36_{\pm 7}$ | $75_{\pm 3}$ |
| DMoN | $78_{\pm 5}$ | $82_{\pm 11}$ | $37_{\pm 7}$ | $76_{\pm 4}$ |
| ECPool | $73_{\pm 5}$ | $84_{\pm 12}$ | $35_{\pm 8}$ | $74_{\pm 5}$ |
| Graclus | $73_{\pm 4}$ | $82_{\pm 12}$ | $33_{\pm 7}$ | $73_{\pm 4}$ |
| $k$-MIS | $75_{\pm 3}$ | $83_{\pm 10}$ | $33_{\pm 8}$ | $73_{\pm 5}$ |
| MinCutPool | $78_{\pm 5}$ | $81_{\pm 12}$ | $34_{\pm 9}$ | $77_{\pm 5}$ |
| Top-$k$ | $72_{\pm 5}$ | $82_{\pm 10}$ | $29_{\pm 7}$ | $74_{\pm 5}$ |
| MaxCutPool | $77_{\pm 4}$ | $84_{\pm 10}$ | $31_{\pm 6}$ | $74_{\pm 4}$ |
| MaxCutPool-E | $77_{\pm 3}$ | $85_{\pm 9}$ | $34_{\pm 5}$ | $74_{\pm 4}$ |
| MaxCutPool-NL | $74_{\pm 4}$ | $83_{\pm 11}$ | $31_{\pm 4}$ | $70_{\pm 4}$ |

## E.2  NODE CLASSIFICATION

Tab. 13 presents the results for node classification using the architecture with skip connections described in Appendix C.4. For this architecture, we focused on the non-expressive variant of MaxCutPool, which consistently delivered superior performance in this context. The improved results can be attributed to the architecture's ability to preserve the original node information through skip connections. Additionally, by avoiding the combination of neighboring node features (as is done in the expressive variant), the model is better equipped to learn high-frequency features, which is particularly advantageous for heterophilic datasets. For the Minesweeper dataset, we chose to use a GIN layer with 16 units as the MP layer, instead of the usual 32 units. This decision was made because, regardless of the pooling method used, the architecture with skip connections consistently achieved nearly 100% ROC AUC whenever configured with a higher capacity.

Table 13: Node classification accuracy (Roman-empire, Amazon-ratings) and AUROC (Minesweeper, Tolokers, Questions) obtained when using the architecture with skip connections.

| Pooler | Roman-e. | Amazon-r. | Minesw.* | Tolokers | Questions | Score |
|---|---|---|---|---|---|---|
| Top-$k$ | $20_{\pm 11}$ | $49_{\pm 7}$ | $91_{\pm 1}$ | $96_{\pm 0}$ | $70_{\pm 3}$ | 1 |
| $k$-MIS | $19_{\pm 2}$ | $53_{\pm 3}$ | $90_{\pm 0}$ | $91_{\pm 2}$ | $82_{\pm 4}$ | 2 |
| NDP | $19_{\pm 4}$ | $56_{\pm 5}$ | $94_{\pm 0}$ | $90_{\pm 8}$ | $69_{\pm 7}$ | 2 |
| MaxCutPool | $67_{\pm 2}$ | $53_{\pm 1}$ | $92_{\pm 1}$ | $96_{\pm 1}$ | $82_{\pm 2}$ | 3 |

# F COMPLEXITY

We first discuss the algorithmic complexities and then report empirical measurements about processing time and memory usage. All measurements are done on an Nvidia RTX A6000.

## F.1 ALGORITHMIC COMPLEXITY

The complexity of MaxCutPool depends on the complexities of the operations SEL, RED, and CON and of the auxiliary loss $\mathcal{L}_{\text{cut}}$.

**SEL** The complexity of the SEL operation depends on the ScoreNet, which consists of a stack of $L$ HetMP layers followed by an MLP, and on the top$_K$ selection.

- **HetMP.** Following the analysis in Blakely et al. (2021), for a graph with $N$ nodes, $E$ edges, and $F$ features, each HetMP layer has a time complexity of $\mathcal{O}(NF^2 + EF)$ and a space complexity of $\mathcal{O}(E + NF + F^2)$. This results in a space and time complexity of $\mathcal{O}(N + E)$ with respect to the input.
- **MLP.** The MLP has a fixed structure with predetermined layer sizes. Since it operates independently on each node's feature vector and the number of operations per node is constant, processing each node takes $\mathcal{O}(1)$ time. With $N$ nodes to process, this results in a total time complexity of $\mathcal{O}(N)$ with respect to the input. The space complexity is also $\mathcal{O}(N)$, as we need to store the MLP hidden and output features for each node.
- **top$_K$** The complexity of sorting an array of $N$ elements is $\mathcal{O}(N \log(N))$. However, if we are interested in finding only the top-$K$ elements, the complexity can be lowered to $\mathcal{O}(N \log(K))$ or $\mathcal{O}(N + K)$, depending on the algorithm adopted. Therefore, we can assume an almost-linear complexity in time and space with respect to the number of nodes $N$.

**RED** For the non-expressive variant, RED involves a Hadamard product between the scores and features of the $K$ selected nodes, giving a time complexity of $\mathcal{O}(K)$. The expressive variant requires an additional multiplication with the assignment matrix $\boldsymbol{S}$, increasing the complexity to $\mathcal{O}(K + NF)$. When $K$ is a function of $N$ (e.g., $K = N/2$), both variants have a time complexity of $\mathcal{O}(N)$ with respect to the input.

The space complexity is $\mathcal{O}(N)$, representing the storage of input and output data.

**CON** Our efficient implementation of the nearest neighbor assignment follows the complexity of BFS: $\mathcal{O}(N + E)$ time and $\mathcal{O}(N)$ space.

**Auxiliary loss** The auxiliary loss $\mathcal{L}_{\text{cut}}$ requires computing a quadratic form, with time complexity $\mathcal{O}(E)$ and space complexity $\mathcal{O}(N + E)$.

**Total complexity** The overall complexity of MaxCutPool is:

$$
\begin{aligned}
\text{Time complexity: } & \mathcal{O}(E + N) \\
\text{Space complexity: } & \mathcal{O}(E + N)
\end{aligned}
\tag{7}
$$

These sub-quadratic complexities match those of the most efficient MP and trainable pooling operators.

## F.2 EXECUTION TIMES

In Tab. 14 we report the number of seconds used by the architecture for node classification to process a batch when configured with different pooling operators.

We note that one-over-$K$ methods such as Graclus, NDP, and $k$-MIS perform a preprocessing step on the CPU before the training starts. Such operations are not accounted for in the measurements in Tab. 14, but they can take significant time and be a bottleneck in those cases that require operations such as eigenvalue decomposition.

Table 14: Execution times in terms of batches processed per second (b/s) by the architecture for node classification configured with different pooling methods.

| Pooler | Roman-e. | Amazon-r. | Minesw. | Tolokers | Questions |
|---|---|---|---|---|---|
| Diffpool | 0.72 b/s | 0.93 b/s | 0.05 b/s | 0.11 b/s | OOM |
| DMoN | 0.66 b/s | 0.83 b/s | 0.06 b/s | 0.11 b/s | OOM |
| MinCutPool | 1.32 b/s | 1.63 b/s | 0.14 b/s | 0.23 b/s | OOM |
| Top-$k$ | 0.01 b/s | 0.01 b/s | 0.01 b/s | 0.01 b/s | 0.03 b/s |
| Graclus | 0.01 b/s | 0.01 b/s | 0.01 b/s | 0.01 b/s | 0.01 b/s |
| $k$-MIS | 0.01 b/s | 0.01 b/s | 0.04 b/s | 0.01 b/s | 0.01 b/s |
| NDP | 0.01 b/s | 0.01 b/s | 0.00 b/s | 0.01 b/s | 0.01 b/s |
| MaxCutPool | 0.03 b/s | 0.10 b/s | 0.01 b/s | 0.09 b/s | 0.13 b/s |

### F.3 MEMORY USAGE

Table 15: Average and maximum GPU memory usage (in MB) by the architecture for node classification when configured with different pooling methods.

| Pooler | Roman-e. | | Amazon-r. | | Minesw. | | Tolokers | | Questions | |
|---|---|---|---|---|---|---|---|---|---|---|
| | Avg. | Max | Avg. | Max | Avg. | Max | Avg. | Max | Avg. | Max |
| Diffpool | 7167.3 | 11277.2 | 8367.8 | 13165.6 | 1397.4 | 2199.4 | 1931. | 3039.7 | OOM | OOM |
| DMoN | 5301.9 | 7359.5 | 6189.5 | 8591.2 | 1035.1 | 1438.2 | 1429. | 1984.7 | OOM | OOM |
| MinCutPool | 7167.8 | 11277.6 | 8367.9 | 13165.8 | 1398.0 | 2200.3 | 1932. | 3040.0 | OOM | OOM |
| Top-$k$ | 2.8 | 3.9 | 3.4 | 4.9 | 1.4 | 1.8 | 3.4 | 6.0 | 5.1 | 8.9 |
| Graclus | 2.5 | 2.6 | 4.0 | 4.1 | 1.5 | 1.5 | 14.6 | 15.1 | 10.4 | 10.6 |
| $k$-MIS | 1.3 | 1.3 | 0.7 | 0.8 | 0.3 | 0.3 | 2.4 | 2.5 | 5.2 | 5.3 |
| NDP | 1.8 | 2.5 | 1.8 | 9.7 | 0.6 | 2.5 | 2.4 | 70.8 | 2.4 | 26.1 |
| MaxCutPool | 13.7 | 25.7 | 16.8 | 31.2 | 6.9 | 12.7 | 27.9 | 52.2 | 32.6 | 61.6 |

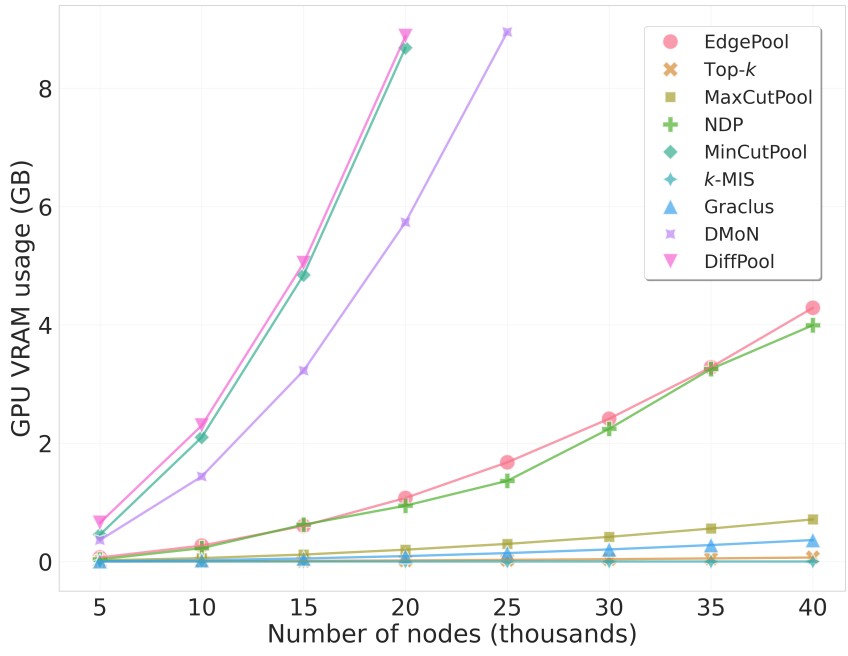

Figure 11: The GPU VRAM usage of the different poolers.

In Tab. 15 we report the average and maximum GPU VRAM used by the architecture for node classification on the different datasets. As for the time, the values reported for Graclus, NDP, and $k$-MIS do not include the SEL and CON operations performed in preprocessing.

To give a more interpretable demonstration of how the space complexity scales for the different pooling methods, in Fig. 11 we report the GPU VRAM usage of the different poolers when processing a randomly generated Erdős-Renyi graph of increasing size. All the graphs are generated keeping $0.01$ as the probability of having an edge between any pair of nodes.

The plot shows that in soft-clustering methods, the GPU VRAM usage grows exponentially with the graph size. On the other hand, for scoring-based methods, including the proposed MaxCutPool, the growth is sublinear, making these approaches extremely suitable for working with large graphs.

