# OpenReview forum: "MaxCutPool: differentiable feature-aware Maxcut for pooling in graph neural networks"
_ICLR.cc/2025/Conference — ICLR 2025 Poster_

### Official Review · Reviewer_Unwn · 2024-10-28

**Soundness:** 3
**Presentation:** 3
**Contribution:** 1
**Rating:** 3
**Confidence:** 5

**Summary:**

This paper proposes a novel graph pooling method based on the maxcut algorithm. As MaxCut assigns high scores to disconnected nodes, this paper motivates its connection to one-every-K pooling methods and score-based methods. They formulate a differentiable form of MaxCut which they integrate into a pooling layer that utilizes heterophilic message-passing layers. Several experiments comparing MaxCutPool to other pooling methods are conducted.

**Strengths:**

The paper proposes an intuitive pooling method. It is very nicely written and nicely describes the current state-of-the-art for graph pooling. I understood almost all of the details.

**Weaknesses:**

* W1: Auxiliarly loss: As with other top k pooling approaches, the selction is a discrete action and thus not differentiable. Optimizing $\mathcal{L}_{cut}$ may not lead to an s that is well-suited for the downstream task. Claiming that the proposed pooling operator is fully differentiable is thus misleading.
* W2: This paper proposes multiple parts without thoroughly analysing each of them:
    * i) Gradient descent for combinatorial problems are typically not preferred as the solution can arbitrarily bad due to local minima. As this paper proposes to use gradient descent for MAXCUT, there needs to be a more detailed comparison with existing methods, ideally in terms of time/space complexity, expected performance and guarantees.
    * ii) It also remains unclear whether this gradient based approach is better for downstream-tasks. The only experiment that compares the proposed MaxCutPool with existing MAXCUT algorithms is in Table 2, which shows slight improvements. To me it is not convincing to use this gradient based approach vs. traditional algorithms within MaxCutPool.
    * iii) While the nearest neighbor aggregation is proposed as a algorithms for many pooling methods, it is only evaluated in combination with MaxCutPool. As the difference is rather small it is unclear whether this step is needed. If the authors decide to propose such an algorithm, a more detailed evaluation would be needed.
* W3: The overall contribution is quite limited. As pointed out by the authors, many algorithms for graph pooling exist and the theoretical and empirical benefits of MaxCutPool do not convince me.
* W4 Experiments:
    * i) There is no reproducible implementation provided.
    * ii) Graph classification: There seems to be a large hyperparameter optimization ofr MaxCutPool, while "all [other] pooling layers were used with the default hyperparameters". This does not seem to be a fair comparison. Especially when only spliiting into train and validation, the search space needs to be of similar size.
    * iii) Heterophilic tasks are only defined for node classification tasks as heterophily is typically defined as having mostly different labels between adjacent nodes. As node labels do not exist for graph-level tasks, it is unclear what heterophilic graph classification means. The authors also did not define what heterophilic graphs are.

Additional minor points:
l.156: Sharpening propagation operators cannot learn any kind of gradients, but sharp gradients.
l.425: "This is the first known instance of a non-expressive pooler passing the expressiveness test provided by this dataset, serving as a counterexample to Theorem 1 in Bianchi & Lachi (2023)". To my understanding, Theorem 1 provides a sufficient condition not a necessary one.

**Questions:**

* l.289: Is CON always defined using the assignment matrix $\mathbf{S}$ based on nearest neighbor aggregation?
* How well does the differentiable MaxCut approximate MaxCut compared to more other algorithms in terms of approximation and time complexity?
* Is the differentiable MaxCut required or does this method work with other algorithms for MaxCut?
* How important is the nearest neighbor aggregation?
* How do the other methods perform with the same level of hyperparameter tuning?
* Why is there no reproducible implementation provided?

---

> ### Author Response · Authors · 2024-11-18
> **Rebuttal (1/2)**
>
> Dear reviewer Unwn,
>
> thanks for the review!
>
> We are afraid there has been a serious misunderstanding that might have put our work under the wrong perspective.
> We try our best to address this incomprehension in the following and we remain available for further clarifications.
>
> **W1**
> The definition of differentiability refers specifically to how gradients flow through the score computation and auxiliary loss, not the top-$K$ selection itself.
> As described in the methods section, our Hadamard product with the score vector $\mathbf{s}$ in the $\texttt{RED}$ operation is either
>
> $\mathbf{X}' = \mathbf{s_i} \odot [\mathbf{X}]_{\mathbf{i}}$
>
> or
>
> $\mathbf{X}' = \mathbf{s}_{\mathbf{i}} \odot \mathbf{S}^\top\mathbf{X}$
>
> This ensures that gradients can flow back through the ScoreNet during backpropagation, thus making our architecture fully differentiable. This is common practice for the top-$K$ pooling operators, as shown e.g., in
> - https://ieeexplore.ieee.org/document/9432709
> - https://arxiv.org/abs/1811.01287
> - https://arxiv.org/abs/1905.02850
>
> This allows the model to learn an effective scoring function that is directly influenced by both the auxiliary loss, which acts as a regularization term, and the downstream task.
>
> We also note that such a differentiability is a key difference between our method and existing algorithms to compute the MAXCUT. It is also what makes it possible to adapt the MAXCUT solution to the content of the node features and to other downstream task losses.
>
> **W2-1, W2-2, W3, Q2, Q3**
> It seems that these comments share a common underlying incomprehension, which we try to clarify in the following.
>
> The reason why we developed MaxCutPool was not to outperform traditional MAXCUT algorithms in terms of cut size, but rather to create a differentiable, scalable, and feature-aware pooling operator that can be trained end-to-end within a GNN.
>
> The results in Table 2, where our gradient-based approach achieves comparable (even superior!) performance to specialized MAXCUT algorithms, serve primarily to validate that our architecture and auxiliary loss effectively capture the essence of the MAXCUT objective. This is particularly noteworthy given that our method lacks the theoretical guarantees of traditional algorithms and, as noted by the reviewer, could potentially get stuck in local minima. The fact that we achieved such strong results demonstrates that
> our model achieves its purpose in implementing an effective maxcut-based regularization objective to perform pooling in a GNN.
>
> We note that it would not be possible to use traditional MAXCUT algorithms within MaxCutPool, because they are not differentiable, they cannot account for node features, and cannot be trained end-to-end with the rest of the GNN.
> Moreover, while traditional MAXCUT algorithms focus solely on optimizing the cut size, our approach can balance this objective with other learning goals by combining multiple losses. This flexibility is essential in a data-driven setting, where the optimal pooling strategy might need to prioritize preserving certain feature patterns or structural properties over maximizing the cut size. The empirical results show that our method, when needed, can achieve high-quality cuts while simultaneously learning to pool nodes in a way that benefits the downstream task.
>
> On the other hand, established MAXCUT algorithms like Goemans-Williamson that offer approximation guarantees, cannot adapt to these task-specific requirements or incorporate node feature information in computing the solution.
> In addition, the theoretical guarantees about the MAXCUT solution are less relevant in downstream tasks such as graph or node classification, where the MAXCUT objective serves only as a regularizer.
>
> **W2-3**
> The improvement in the expressive power of any score-based pooling operator that adopts the proposed aggregation scheme is guaranteed by the theoretical result on the expressiveness of pooling operators (https://arxiv.org/abs/2304.01575).
> As such, our claim does not require the support of an experimental evaluation.
>
> **W4-1, Q6**
> We created an anonymous version of our repository, which can be accessed at https://anonymous.4open.science/r/MaxCutPool/.
>
> **W4-2, Q5**
> We believe there is a misunderstanding here. We did not just use a train-validation split. Instead, we used a rigorous 10-fold cross-validation procedure where the training set was further split into 90-10\% train-validation sets.
>
> Regarding the hyperparameters optimization, as shown in the Appendix, the hyperparameter search space for MaxCutPool was actually quite constrained, resulting in a relatively small grid of configurations compared to the complexity of the model. Most of the other pooling methods do not have hyperparameters, except for the soft-clustering methods. For them, we rely on the thorough hyperparameters optimization performed by their respective authors on the different datasets, which we refer to as "default" values.

---

> > ### Author Response · Authors · 2024-11-18
> > **Rebuttal (2/2)**
> >
> > **W4-3**
> > We provided those definitions in the Appendix. However, since it was not mandatory to check the Appendix, we perfectly understand that they could have been overlooked.
> >
> > The surrogate homophily score $\bar{h}(\mathcal{D})$ that we provided in section D.3 of the appendix and the homophily score $h(\mathcal{D})$ referenced in D.4 should address the concern of the reviewer.
> > We remain available for further clarification.
> >
> > **minor1**
> > Thanks for the correction, we changed the manuscript accordingly.
> >
> > **minor2**
> > That is correct. What we meant was that we provide a first example to show that the condition is, indeed, only sufficient and not necessary. In fact, the paper from Bianchi \& Lachi does not provide an example of a non-expressive pooling operator that achieves perfect accuracy on the EXPWL1 dataset.
> >
> > We removed that expression containing the word "counterexample" to avoid confusion.
> >
> > **Q1**
> > Yes, that's correct.
> >
> > **Q4**
> > Without it would not be possible to form an adjacency matrix for the pooled graph that is both sparse and not disconnected.
> > The nearest neighbor aggregation emerges as a natural way to handle nodes that aren't selected as supernodes. Once we've identified representative nodes through the MAXCUT-based selection, we need to decide how to handle the remaining nodes. Assigning each node to its closest supernode (in terms of graph distance) is a reasonable approach that preserves local graph structure and sparsity, and ensures connectivity.

---

> > > ### Comment · Reviewer_Unwn · 2024-11-19
> > >
> > > I thank the reviewer for their rebuttal. Unfortunately, none of my major concerns were addressed.
> > >
> > > ### Differentiability
> > > The authors propose two ways for differentiation, none of which provide meaningful gradient information for optimization. The authors not being aware of this large limitation is a major concern in itself.
> > >
> > > #### Top-K
> > > It is proposed to multiply the supernodes by the scores of the selected nodes using $\mathbf{X}^\prime = \mathbf{s}_i \odot [\mathbf{X}]_i$ or $\mathbf{X}^\prime = \mathbf{s}_i \odot \mathbf{S}^T\mathbf{X}$.While researchers proposed this more than five years ago because this was not better understood, there are large communities researching on these topics and a lot of progress has been made. See, e.g., [1,2,3]. There are no meaningful gradients flowing through this operation. Multiplying scores with the node states does not provide gradient information on whether other nodes should have been selected. Claiming that this operation makes the model fully differentiable is strongly misleading.
> > >
> > > #### Auxiliary Loss
> > > The issue here is similar. While this MaxCut is differentiable, it is task-independent, so the model remains as not end-to-end differentiable. As such, the claims that this MaxCut can be adapted to downstream tasks are not supported. For the optimization of any given task, it would not matter if this differentiable MaxCut is utilized or if any traditional MaxCut algorithm was employed. If the Top-K selection was differentiable, this differentiable MaxCut could indeed serve as a regularization.
> > >
> > > >**W2-3** As such, our claim does not require the support of an experimental evaluation.
> > >
> > > This seems weird to me. While this nearest neighbor assignment is applied to MaxCutPool, it is not applied to other methods considered, such as Top-k pool. As such, it is unclear whether the performance improvements result from this step or other steps. It could be that Top-k pool gets similarly improved by such an assignment or that it is empirically not relevant.
> > >
> > > Overall, I think that this method still requires a lot of work before it can be published.
> > >
> > > ---
> > > [1] Petersen et al., Differentiable Top-k Classification Learning, ICML (2022).
> > >
> > > [2] Xie et al., Differentiable top-k with optimal transport, NeurIPS (2020).
> > >
> > > [3] Cordonnier et al., Differentiable patch selection for image recognition, CVPR (2021).

---

> > > > ### Author Response · Authors · 2024-11-19
> > > >
> > > > We sincerely thank you for having taken the time to answer us and for engaging in this scientific discussion!
> > > >
> > > > # Top-k & gradients
> > > > Your criticism seems to be addressed more toward the top-$K$ operation, which is very standard in graph pooling, rather than to our contribution.
> > > > To give you an idea of the widespread use of top-$K$, please consider how many modern pooling operators are adopting it, e.g., by checking Table 2 here (https://arxiv.org/abs/2204.07321) or Section 3.2.2 here (https://link.springer.com/article/10.1007/s10462-024-10918-9). If top-$K$+gating would not provide meaningful gradient information, all these methods could not work.
> > > >
> > > > We believe that the attention should be directed not to the top-$K$ operation itself, but rather to how the score vector $\mathbf{s}$ is generated and utilized. This score is learned through an MLP that processes **all** nodes' embeddings. It's true that gradients do not flow through the discrete top-$K$ selection.
> > > > However, thanks to the Hadamard product, which can be seen as a skip/gating operation, **they do flow** through the values in $\mathbf{s}$ and, thus, to the MLP and the MP layers.
> > > > This allows the network to learn which nodes should receive high scores based on both the auxiliary loss and the downstream task, which can each all the trainable weights of the GNN.
> > > >
> > > > ## Further evidence
> > > > Please, check this plot of the gradients of a simple GNN classifier consisting of GIN-MaxCutPool-GIN-readout: https://imgur.com/a/gradient-flows-maxcutpool-2gnzUiz
> > > >
> > > > The 3 plots show the GNN trained with:
> > > > 1. task + aux loss
> > > > 2. only task loss
> > > > 3. only aux loss
> > > >
> > > > As you can see, the only weights that get no gradients are those of the MP after the pooling layer when the GNN is trained only with the auxiliary loss, which is expected.
> > > >
> > > > Please, consider also our ablation study with the MaxCutPool-NL variant, which removes the auxiliary MAXCUT loss and relies solely on the downstream task loss.
> > > > It shows that our model is still able to learn meaningful scores guided only by the task objective. While this variant generally performs worse than the full model (confirming the importance of our MAXCUT regularization), its ability to achieve competitive results proves that task-specific loss effectively influences the score generation process through backpropagation.
> > > >
> > > > ## Future work
> > > > With that said, all the non-differentiable operations used in deep learning can create discontinuous changes in the gradients, which can pose difficulties during training. This is something well-known and the papers that you linked propose solutions to this.
> > > > The idea of incorporating a differentiable top-$K$ in graph pooling would certainly be an interesting direction to explore. However, there are many aspects to account for when porting these methodologies to a GNN. For example, the soft top-$K$ proposed here (https://arxiv.org/abs/2104.03059) would make the pooled graph dense, the optimal transport here (https://arxiv.org/abs/2002.06504) has a cubic complexity and, thus, not suitable for GNNs... And so on.
> > > > Most importantly, we believe that this would be a contribution of general interest for all score-based methods, but it's quite orthogonal to the contribution of this paper.
> > > >
> > > > # Auxiliary loss
> > > > We respectfully think that the statement *"While this MaxCut is differentiable, it is task-independent"* is incorrect. The MAXCUT loss depends on the score vector $\mathbf{s}$, which is computed from the MP layers and the MLP. Since the weights of the MP and MLP layers are, in turn, influenced by the downstream task, we can conclude that the MaxCut is, in fact, task dependent.
> > > >
> > > > On the other hand, a traditional MaxCut algorithm would have looked only at the topology of the input graph. As such, it would have given the same cut every time and, thus, the same retained nodes without being affected neither by the node features nor by the downstream task.
> > > >
> > > > # W2-3
> > > > We agree that it is not possible to say *a-priori* if a generic top-$K$ improves its performance with the nearest neighbor assignment.
> > > > However, that was not our claim.
> > > > As we state in our paper: *"Our scheme can be applied not only to our method but to the whole family of sparse scoring-based graph pooling operators enhancing, in principle, their representational power."*
> > > > The enhanced expressiveness is grounded in theoretical guarantees and is independent of empirical performance.

---

> > > > > ### Comment · Reviewer_Unwn · 2024-11-22
> > > > >
> > > > > > If top-k+gating would not provide meaningful gradient information, all these methods could not work.
> > > > >
> > > > > Top-k is a great way to do pooling and I am not saying that selecting nodes based on MinCut or something else does not make sense or cannot work. It just does not provide any meaningful gradients to the scores. Multiplying the scores with the output does not make the selection differentiable. This is not a discussable point.
> > > > >
> > > > > In Top-k pooling, the authors are fully aware of this and only claim that the gating scores allow gradients to flow into the score calculation. Claiming that this would be fully-differentiable or end-to-end learning is unacceptable. If the authors changed their claims and stated this non-differentiability of the selection as a weakness, the claims would at least be correct.
> > > > >
> > > > > > This allows the network to learn which nodes should receive high scores based on both the auxiliary loss and the downstream task, which can reach all the trainable weights of the GNN.
> > > > >
> > > > > Reaching all trainable weights is not relevant. Maximizing the scores directly would also reach all trainable weights. However, this does not allow learning which of these scores are relevant for the task.
> > > > >
> > > > > > We can conclude that the MaxCut is, in fact, task dependent.
> > > > >
> > > > > This is only true if the selection would be differentiable. In its current form, it's not.
> > > > >
> > > > >
> > > > > The authors can see the hurtfulness of such misleading claims by their own statements. When other researchers get convinced that this Top-k selection would be a differentiable operation, new methods (like MinCutPool) get proposed that build on a false basis. Accepting such a paper would be hurtful to the community and set back research on graph pooling as more researchers could follow this idea. Therefore, I increase my confidence in rejecting this paper.

---

> > > > > > ### Author Response · Authors · 2024-11-22
> > > > > >
> > > > > > Dear unwn,
> > > > > >
> > > > > > thanks again for your feedback and for taking part in this discussion!
> > > > > >
> > > > > > We are glad to hear that you recognize that Top-k is a great way to do pooling, that our approach makes sense, and it can work. The main issue seems to be about the terminology, that we address below.
> > > > > >
> > > > > > # We never claimed that top-$K$ is differentiable
> > > > > > Testament to that are the following excerpts from our paper:
> > > > > >
> > > > > > > "The $\texttt{RED}$ operation computes the node features of the pooled graph $\boldsymbol{X}'$ by multiplying the features of the selected nodes $\boldsymbol{X}_i$ with the scores $\boldsymbol{s}$. This operation is necessary to let the gradients flow past the top$_K$ operation, which is not differentiable."
> > > > > >
> > > > > > and
> > > > > >
> > > > > > > "The Hadamard product $\odot$ enables gradients flowing through the ScoreNet during back-propagation, making the model fully differentiable despite the top$_K$ operation."
> > > > > >
> > > > > > However, we concede that in the latter sentence "fully differentiable" is not precise and we changed into:
> > > > > > > "The Hadamard product $\odot$ enables gradients flowing through the ScoreNet during back-propagation, making it possible for the gradients of the task loss to reach every component of our model, despite the non-differentiable top$_K$ operation."
> > > > > >
> > > > > > We also modified other parts of the paper where we claimed that our model is "fully differentiable".
> > > > > >
> > > > > >
> > > > > > # End-to-end training and "non meaningful gradients"
> > > > > >
> > > > > > If by "meaningful gradients" you mean the gradients flowing directly through the node selection process (like in the differentiable top$_K$ references you provided)
> > > > > > we **respectfully disagree** that it's the only option for end-to-end learning.
> > > > > > Many successful deep learning approaches operate through *indirect* gradient pathways, particularly when dealing with discrete operations.
> > > > > > Consider, e.g., policy gradient methods in reinforcement learning: despite having a non-differentiable action sampling step, these methods successfully learn complex discrete decision-making policies by optimizing the parameters that generate action probabilities.
> > > > > >
> > > > > > Our approach follows the same principle: while the top-K selection is indeed non-differentiable, ScoreNet learns effective scoring policies through:
> > > > > >
> > > > > > - A direct pathway where the auxiliary MAXCUT loss provides gradients to optimize the cut size.
> > > > > > - An indirect pathway where the task loss influences score generation through the Hadamard product in the REDUCE operation.
> > > > > >
> > > > > > When the task loss indicates that certain nodes should be preferred, the gradients flowing through the Hadamard product update ScoreNet's parameters, leading to different scores and thus different selections in subsequent forward passes.
> > > > > > This is exactly like policy networks in RL that learn to adjust action probabilities based on reward signals.
> > > > > >
> > > > > > Based on your point of view, all the RL architectures are also not end-to-end trainable? All the policy gradients in these models are "not meaningful"? If yes, how can they work?
> > > > > >
> > > > > > However, if we misunderstood your concern about "meaningful gradients", we would greatly appreciate it if you could provide a formal definition of what constitutes a meaningful gradient in your view, so we can better address your specific concerns.

---

> > > > > > > ### Comment · Reviewer_Unwn · 2024-11-25
> > > > > > >
> > > > > > > Terminology is the first important step so that the claims are not wrong anymore.
> > > > > > >
> > > > > > > Even more important are the issues regarding the functionality.
> > > > > > >
> > > > > > > ### Meaningful gradients
> > > > > > > By meaningful gradient I mean that taking a step of gradient descent based on the target loss actually reduces the target loss as required for convergence guarantees of gradient descent (see, e.g., [1]). Policy gradient methods are indeed a great example:
> > > > > > >
> > > > > > > > Based on your point of view, all the RL architectures are also not end-to-end trainable? All the policy gradients in these models are "not meaningful"? If yes, how can they work?
> > > > > > >
> > > > > > > Policy gradient is completely different from the considered selection. They estimate the true gradient by stochastically selecting elements based on a probability distribution that gets updated by how that selection affects the results. Please take a look at the policy gradient theorem [2]. There are convergence guarantees that gradient descent can find good solutions.
> > > > > > >
> > > > > > > In your considered Top-K selection, the initially selected nodes are task-independent (they correspond to the smallest eigenvector given the HetMP). If, for a given task, selecting different nodes would lead to better results, how can such optimization happen? You cannot increase the score of specific nodes you have not selected, as there is no gradient for the score calculation of these nodes. How should gradient descent decrease the score of individual selected nodes? There are no gradients that contain the information on which node selections improved or worsened the results. As such, gradient descent cannot identify which nodes should be selected in order to reduce the target loss.
> > > > > > >
> > > > > > > Thus, I don't see the advantage of using a differentiable MaxCut. Any other MaxCut algorithm should work similarly, as classical methods also do not get any relevant task information. Thus, either MaxCutPool should be defined independently of the MaxCut algorithm or, if the authors are convinced of the benefits of a differentiable MaxCut, it should be clearly evaluated that this choice is beneficial.
> > > > > > >
> > > > > > > ---
> > > > > > > [1] Garrigos & Gower, Handbook of Convergence Theorems for (Stochastic) Gradient Methods, 2023.
> > > > > > >
> > > > > > > [2] Sutton et al., Policy Gradient Methods for Reinforcement Learning with Function Approximation, NeurIPS (1999).

---

> > > > > > > > ### Author Response · Authors · 2024-11-26
> > > > > > > >
> > > > > > > > Dear Reviewer Unwn,
> > > > > > > >
> > > > > > > > Please, find below an answer to the concerns raised in your last response.
> > > > > > > >
> > > > > > > > > "Any other MaxCut algorithm should work similarly, as classical methods also do not get any relevant task information."
> > > > > > > >
> > > > > > > > Testing the differences between our method and classical algorithms for MaxCut to perform pooling was the first thing we investigated. The positive results we obtained were the premise that motivated us to pursue this research direction.
> > > > > > > >
> > > > > > > > As further and final evidence to prove the difference between our approach compared to a standard maxcut algorithm please consider the following figure directly built from our experiments' logs: https://imgur.com/a/evolution-of-scores-node-selection-maxcutpool-Nf1MkMo
> > > > > > > >
> > > > > > > >  > "You cannot increase the score of specific nodes you have not selected, as there is no gradient for the score calculation of these nodes."
> > > > > > > >
> > > > > > > >  In the example reported in the figure, the GNN is the No-Loss (NL) variant, meaning that the model is trained **without** the auxiliary loss and the only gradients are those from the classification loss.
> > > > > > > >
> > > > > > > > The plot clearly shows how the scores and node selections evolve during training on the Multipartite graphs dataset. While a non-learnable algorithm like the largest eigenvector vertex selection (LEVS) produces a fixed assignment based just on topology, our method guided only by the task loss successfully adapts its scores during training, leading to the selection of an independent set of nodes (the orange ones) needed to solve the task.
> > > > > > > >
> > > > > > > > > "if the authors are convinced of the benefits of a differentiable MaxCut, it should be clearly evaluated that this choice is beneficial."
> > > > > > > >
> > > > > > > > Experiment 4.1 focuses exactly on this. Also, experiments 4.3 and the one in Appendix C.4 provide a comparison against NDP, which is an implementation of LEVS for graph pooling.
> > > > > > > >
> > > > > > > > The results are obtained with our publicly available implementation, which can be used to reproduce these results. We welcome any additional experiments that you, or other researchers, might find valuable.
> > > > > > > >
> > > > > > > > # Concluding remarks
> > > > > > > >
> > > > > > > > As the rebuttal period draws to a conclusion, we would sincerely like to thank you for engaging in this discussion, for taking the time to explain your arguments in detail, and for providing us with useful references, which gave us good food for thought.
> > > > > > > > We respect your point, we agree that the lack of stochasticity sets the top-$K$ selection apart from the policy gradients, and we also agree that indirect gradient pathways do not enjoy theoretical guarantees about finding the optimal solution. Nevertheless, our method consistently works and the extensive experimental evaluation to support our claims is rock solid. As such, we believe that our work can be a valuable contribution to the GNN community.

---

> > > > > > > > > ### Comment · Reviewer_Unwn · 2024-12-02
> > > > > > > > > **Final Response**
> > > > > > > > >
> > > > > > > > > I thank the authors for their answers and interesting explanations. I extensively looked into various aspects of the paper and agree that some of the ideas have potential. Unfortunately, I am not convinced of the benefits of this work in its current form and have decided to maintain my score.
> > > > > > > > >
> > > > > > > > > The example indeed confirms that the masking process is differentiable. However, this is independent of the pooling operator, i.e., keeping all nodes and performing the operation $\mathbf{X}_i = s_i \mathbf{X}_i$ would likely show the same behavior. The difference between MaxCutPool and Top-k pooling seems to be that MaxCut selects diverse nodes, and Top-k likely selects nodes from the same cluster. As these poolings are not differentiable, Top-k can only solve the task when the initial selection already contains at least some of the required nodes. I do agree that this diverse selection is an advantageous property of MaxCutPool. At the same time, it questions the need for the non-differentiable pooling operator in addition to the mask.
> > > > > > > > >
> > > > > > > > > As described above, the experiments do not convince me to use this method.
> > > > > > > > >
> > > > > > > > > For Experiment 4.1, the authors state in l.1066: "The unpooling strategy used in this architecture is the broadcast one for k-MIS and MaxCutPool and the padding one for Top-k and NDP". Thus, the results do not seem comparable. When showing the advantages of some capability, it needs to be evaluated independently of the other parts. For example, by performing a grid search over the unpooling strategy. Dedicated studies on other parts would have also been more insightful, e.g., using the max eigenvector for top-k selection while keeping all other parts equal to MaxCutPool, including the mask.
> > > > > > > > >
> > > > > > > > > In Table 13, some of the results seem comparable as the same unpooling strategy was used for Top-k, NDP, and MaxCutPool. MaxCutPool is the best-performing method on Roman-Empire, which is good. However, the reason behind this is unclear.

---

> > > > > > > > > > ### Author Response · Authors · 2024-12-03
> > > > > > > > > >
> > > > > > > > > > We acknowledge your final comments, which we address in the following.
> > > > > > > > > > We are, however, disappointed that your evaluation has remained unchanged despite our extensive responses and clear empirical evidence addressing each of your initial concerns. This suggests a predetermined stance rather than an objective evaluation based on the evidence and clarifications provided.
> > > > > > > > > >
> > > > > > > > > > ---
> > > > > > > > > >
> > > > > > > > > > Your comment
> > > > > > > > > >
> > > > > > > > > > > "it questions the need for the non-differentiable pooling operator in addition to the mask"
> > > > > > > > > >
> > > > > > > > > > is very concerning because it overlooks the fundamental purpose of graph pooling. Simply masking node features without applying top-k would not make possible to create the coarsened graphs that are needed to obtain hierarchical representations.
> > > > > > > > > >
> > > > > > > > > > ---
> > > > > > > > > >
> > > > > > > > > > Your proposal to compare with
> > > > > > > > > >
> > > > > > > > > > > "max eigenvector for top-k selection while keeping all other parts equal to MaxCutPool"
> > > > > > > > > >
> > > > > > > > > > is precisely what NDP does: it uses the largest eigenvector of the graph Laplacian for node selection.
> > > > > > > > > > Therefore, the experiment you are asking for is already present in the paper and demonstrates that our learnable approach, which incorporates both structural and task-specific information, consistently outperforms NDP's fixed eigenvector-based selection, particularly on challenging heterophilic graphs like Roman-Empire.
> > > > > > > > > >
> > > > > > > > > > ---
> > > > > > > > > >
> > > > > > > > > > Your comment about unpooling strategies
> > > > > > > > > >
> > > > > > > > > > > "the results do not seem comparable"
> > > > > > > > > >
> > > > > > > > > > overlooks a fundamental point.
> > > > > > > > > > Top-k and NDP **must** use padding because they lack the ability to track where unselected nodes should go. These methods simply discard nodes without assigning them to the selected ones, making the broadcasting strategy structurally impossible. Therefore, this is not a matter of choices in our experimental design but a consequence of an inherent limitation of these competing methods.
> > > > > > > > > >
> > > > > > > > > > With that said, all the experimental details were available throughout the entire review period. We find it concerning that you introduce entirely new criticisms about an additional experiment in the appendix during the final day of the rebuttal period. This seems more focused on finding points of criticism rather than providing constructive feedback that could  advance the scientific discussion.

---

### Official Review · Reviewer_E67D · 2024-11-01

**Soundness:** 2
**Presentation:** 3
**Contribution:** 3
**Rating:** 6
**Confidence:** 4

**Summary:**

Graph neural networks are constructed out of a few relatively simple building blocks -- A propagation operator, a message function, and a pooling function. These three components have received significant attention over the last few years, with both heuristic and graph theoretic approaches to improve the performance of GNNs on a variety of graph ML tasks. In this work, the authors present a novel pooling operator termed MaxCutPool, that is inspired by the classic graph theory problem of finding the max cut. They provide justification for the value of this pooling operator through robust experiments to find significant performance improvements.

**Strengths:**

1. The paper is clearly presented
2. The experiments are well defined and clearly motivated
3. The authors make significant efforts to contextualize their work within the rest of the literature.
4. The algorithm is clearly outlined in enough detail to reimplement

**Weaknesses:**

This paper is well written, but not without weaknesses. In particular:

1. The authors do not discuss the computational cost of running their pooling operators in comparison to others. While potentially less relevant for graph classification tasks, it is highly relevant for node classification tasks. Please include some discussion, at least in the appendix, of this.
2. The authors do not justify _why_ it is expected that the max cut should provide a better pooling operator.
3. The results in experiment 4.2 indicate only mild performance gains, and the presented method frequently underperforms the case with no pooling. It's was hard for me to understand from the text what the intuition for this is.
4. The experiments in section 4.3 are hard to understand without a no-pooling option.

**Questions:**

1. In experiment 4.1, what were the features used for GNN? Were these learnable vertex embeddings?
2. How does maxcutpool scale with the number of vertices in a graph?
3. What is the intuition as to why learning the max cut operator provides a good pooling operator? Is it just that this is a high-frequency projection?
4. In experiment 4.3, could you provide the tuned baseline for a GNN with no pooling?
5. With $\delta$ set to 2 in your node classification experiments, your propagation operator is already tuned towards heterophilic settings, and the stated intuition for why maxcutpool works is because it is projecting out high frequency components. Could you provide a node-classification results as a function of delta? Given that these datasets are all heterophilic, it's hard to tease out whether the improvements are due to the pooling or the propagation operator.
6. How was hyperparameter tuning performed?
7. Can this work be extended to link prediction in any meaningful way? Would it be possible to view super-node membership as a labeling trick?
8. Does the unpooling operator involve any normalization?

---

> ### Author Response · Authors · 2024-11-18
> **Rebuttal (1/2)**
>
> Dear reviewer E67D,
>
> thank you very much for the positive and very detailed review. There are a lot of interesting comments and suggestions that we appreciate.
>
> **Weakness 1**
> We improved the computational complexity analysis and comparison with the other methods. We now included in Appendix F a detailed report about the time and memory consumption of different pooling layers.
>
> **Weakness 2**
> Thanks for bringing up this important point.
>
> Traditional score-based methods are renowned for their efficiency and scalability.
> However, they select nodes based on scores that are obtained from node features produced by homophilic MP operations.
> As such, the selected nodes usually come from the same part of the graph. This is a well-known problem because the rest of the graph is not represented.
>
> On the other hand, selecting the nodes belonging to one side of the maxcut partition provides a much more uniform sampling, ensuring that all the parts of the original graph are represented after pooling.
> In addition, when applying standard homophilic MP layers before MaxCutPool, the stronger the association between a pair of nodes,
> the more similar their features will be. Keeping them both will, thus, be redundant, and one of them can be dropped. This is precisely what is done by MAXCUT, which selects nodes with few direct links.
>
> We inserted this explanation in the new section 3.2 of the revised paper.
>
> **Weakness 3**
> The "No pool" baseline is used to establish whether pooling in general can improve performance on a given dataset, which is not always the case. This is a rather standard procedure in graph pooling evaluations.
> For example, if the graphs are too small, using graph pooling can lower the performance w.r.t. the baseline.
> Nevertheless, in those cases, it is still meaningful to compare different pooling operators to evaluate their capability of retaining enough information. That is why the ranking only focuses on those comparisons.
>
> Given this context, the results consistently demonstrate that our method outperforms existing approaches, with statistically significant improvements across the evaluation metrics. In particular, statistical analysis (ANOVA followed by Tukey-HSD tests, both with p-value 0.05) revealed that our method is consistently equal or superior to every other pooling operator.
>
> **Weakness 4**
> We added the "No pool" baseline to Table 4.
>
> We would like to point out that this experiment is inspired by the test "Preserving node attributes" from section 5 in the SRC paper (https://arxiv.org/abs/2110.05292). There, the goal is to check the capability of a pooling method to retain as much information as possible from the input data.
> As such, and similarly to the graph classification experiment, the "No pool" should be intended as a baseline to provide context rather than a competitor.
>
> **Question 1**
> Some of the PyGSP graphs had 2-D coordinates used for plotting that we used as surrogate features. All the remaining graphs had no features and we used a constant value. The details can be found in Appendix C.2.
>
> We would like to comment that the point of this experiment was to show that our model can produce MAXCUT partitions that are comparable, or even better, to those of traditional algorithms specifically designed to compute the MAXCUT.
> Using learnable vertex features would have added additional capacity to our model and arguably improved its performance. However, this would have been different from the setting used for graph and node classification, which is our main focus.
>
> **Question 2**
> Our model scales linearly with the vertices and edges of the graph.
> This experiment is provided in Appendix F, alongside with other computational cost assessments.
>
> **Question 3**
> In addition to the answer provided to Weakness 2, we would like to add the following.
>
> While the MAXCUT objective does promote high-frequency signals through node differentiation, it's more sophisticated than just a high-frequency projection. The MAXCUT auxiliary loss $\mathcal{L}_{cut}$ acts as a regularization term that is optimized jointly with the downstream task loss. This allows the model to find a partition that adaptively balances between preserving distinct node information through the MAXCUT objective and is optimal for task performance. The task loss can even guide the model to ignore the MAXCUT objective when beneficial, as demonstrated by datasets like COLLAB where the auxiliary loss converges to near zero -- instead of its minimum value of -1.
>
> **Question 4**
> Please, refer to our answer to Weakness 4.

---

> > ### Author Response · Authors · 2024-11-18
> > **Rebuttal (2/2)**
> >
> > **Question 5**
> > We believe there is a misunderstanding here. The heterophilic message passing operator is only used within the ScoreNet component of MaxCutPool to compute node scores for selection, not in the main GNN architecture used for node classification. The core GNN uses standard GIN layers for message passing. This design choice ensures that the improved performance on heterophilic graphs can be attributed to the pooling mechanism itself, rather than the propagation operator.
> >
> > We clarified this better in Section 3.2 of the revised manuscript.
> >
> > **Question 6**
> > The hyperparameters were tuned via a grid search in a small set of configurations. The details are provided in Appendix C.
> >
> > **Question 7**
> > Thanks for the interesting suggestion, we will definitely think more about this!
> > However, our first impression is that this idea seems more suitable for soft-clustering approaches such as MinCutPool and Diffpool, that identify clusters of nodes that are similar and densely connected. As such, two nodes that are in the same cluster are more likely to be linked.
> >
> > **Question 8**
> > No, it's just the inverse mapping of the pooling. However, please note that we implemented two different strategies and reported the details in Appendix C4.

---

> > > ### Comment · Reviewer_E67D · 2024-11-27
> > >
> > > Thank you for your thorough and detailed rebuttal, and in particular clarifying my misunderstanding in Q5. I feel as if you've adequately answered the questions.

---

### Official Review · Reviewer_ka5D · 2024-11-04

**Soundness:** 3
**Presentation:** 3
**Contribution:** 3
**Rating:** 6
**Confidence:** 2

**Summary:**

The paper introduces a novel GNN-based approach for solving the MaxCut problem in attributed graphs. The proposed method exhibits good performance across multiple experiments, particularly when evaluated on the newly introduced heterophilic graph benchmark dataset for graph classification tasks.

**Strengths:**

- The writing is generally reader-friendly, featuring detailed illustrations such as figures and hyperparameters. Theoretical analyses are also provided.
- The proposed method is easy to understand yet remains effective, performing well in experiments.
- This work includes comprehensive experiments covering MaxCut partition computation, graph classification, and node classification, which lend credibility to the conclusions presented in the paper.

**Weaknesses:**

Please refer to the questions below.

**Questions:**

* The method uses HetMP, which can be interpreted as a Laplacian sharpening kernel that enhances differentiation between signals across nodes, thereby reducing smoothness. This approach performs well on heterophilic graphs. However, in Figures 2(b-c), the steps seem to be based on a homophily assumption—where neighboring nodes tend to belong to the same cluster. Could you provide further clarification on this?
* Beyond Equation (4), how does the proposed method perform when using other graph kernels or filters, such as high-pass filters? Additionally, how does the method perform on homophilic graphs?
* Can the proposed method scale to graphs of arbitrary size, or are there practical limitations on the scalability threshold?

---

> ### Author Response · Authors · 2024-11-18
> **Rebuttal**
>
> Dear reviewer ka5D,
>
> We sincerely thank you for the review and for the useful comments that pointed out some lack of clarity in our paper. This is much appreciated.
>
> **Question 1**
> Indeed, the supernode selection in $\texttt{SEL}$ is explicitly designed to be heterophilic. This is done to sample the graph *uniformly*, differently from other score-based poolers that select nodes concentrated in a single region because they compute the scores directly from homophilic propagation.
>
> As correctly noted by the reviewer, the assignments $\mathbf{S}$ are computed through the nearest-neighbor aggregation, which follows a homophilic assumption.
> Assignments built in this way synergize well with the uniform sampling of the supernodes and are what allow the pooled adjacency $\mathbf{S}^\top\mathbf{A}\mathbf{S}$ to be connected yet sparse.
>
> A tension between heterophilic and homophilic operations might appear when looking at how the features of the supernodes are computed by $\texttt{RED}$.
> However, it should be stressed that the homophilic assignments *can* be leveraged by $\texttt{RED}$ to ensure expressiveness when summarizing the graph content, but that is not mandatory.
> In fact, for heterophilic datasets, the non-expressive variant of our method offers a more suitable alternative.
> In the latter case, $\mathbf{S}$ is not used and the formation of supernodes is consistent with the heterophilic operation of the HetMP layer.
>
> We clarified this point in the new section 3.2 of the paper.
>
> **Question 2**
> The operator defined in Equation (4) is extremely simple as it is based on a standard GCN, yet it gives us exactly what we need to get a MAXCUT partition.
> Even if other heterogeneous MP layers could have been used, they would have been arguably more complex and bring no additional benefits that we could think of.
>
> Regarding performance on homophilic graphs, the architectures we use for graph and node classification can also handle these cases effectively. The GIN layers that precede the pooling layer perform standard homophilic propagation, allowing the network to identify and leverage similarity patterns when they are present in the data. This is empirically validated by our experimental results, where MaxCutPool achieves competitive performance even on strongly homophilic datasets like GCB-H (reported in the graph classification section) and MUTAG (reported in the Appendix), which have high $\bar{h}(\mathcal{D})$ values of 0.8440 and 0.7082, respectively. In COLLAB (the extreme case of homophily where all the nodes have the same feature) our method adapts by letting the auxiliary loss converge to zero instead of its minimum theoretical value of -1.
>
> **Question 3**
> Our method scales very well, much better than pooling methods that use dense operations. We improved our scalability analysis including more detailed comparisons of time and memory requirements of our pooling layer compared to others. This can be found in Appendix F of the revised manuscript.

---

> > ### Comment · Reviewer_ka5D · 2024-11-22
> >
> > I appreciate your detailed reply. Most of my concerns have been resolved, and I have accordingly raised my score.

---

### Official Review · Reviewer_iy7r · 2024-11-11

**Soundness:** 3
**Presentation:** 3
**Contribution:** 2
**Rating:** 6
**Confidence:** 3

**Summary:**

This paper introduces an innovative method for computing the MAXCUT in attributed graphs which is fully differentiable, enabling joint optimization of the MAXCUT with other objectives. Leveraging the resulting MAXCUT partition, the authors develop a hierarchical graph pooling layer tailored for Graph Neural Networks. The authors claim that the pooling layer is sparse, differentiable, and effective for downstream tasks on heterophilic graphs, addressing a key challenge in graph learning by providing a versatile and adaptable pooling strategy.

**Strengths:**

- Introducing the concept of maxcut into attributed graph pooling is novel and meaningful, and it is worth noting the proposed method is feature-aware and differentiable.
- The background and related work are introduced in detail and accompanied by illustrations, which greatly aid in understanding the proposed method.
- It is encouraging that the proposed method was tested on three tasks, including maxcut partition, graph classification, and node classification.

**Weaknesses:**

- The authors are advised to theoretically and empirically analyze the complexity of MaxCutPool, showing whether it introduces additional computational overhead to GNNs.
- It is also recommended that the authors provide the code to facilitate a better understanding of the proposed method and to ensure the reproducibility of the experiments.
- The authors claim that MaxCutPool is particularly effective for heterophilic graphs, but no detailed analysis is provided. What is the underlying interaction between MaxCutPool and heterophilic graphs? Additionally, considering that traditional message passing (including GIN) is designed for homophilic graphs and is regarded as a low-pass filter, could this conflict with MaxCutPool and potentially lead to poorer results?
- My primary concern is with the experimental section. In graph classification tasks, the proposed approach does not demonstrate satisfactory performance and even falls below the no-pooling baseline on nearly half of the datasets. In Table 3, the authors mark the performance of the no-pooling baseline in gray, even when it ranks highest, which is confusing.
- Is MaxCutPool-NL learning $\mathbf{s}$ solely with task loss after removing $\mathcal{L}_{cut}$? Does this imply that the core concept of MaxCut has been removed? If so, it still appears to be comparable.
- Although the work emphasizes its advantages on heterophilic graphs, it does not perform better than other pooling methods on the GCB-H and MUTAG datasets, which have the highest levels of heterophily.
- Compared to graph classification, the performance on node classification is satisfactory. However, it seems that datasets are not the commonly adopted ones for node classification (such as Planetoid or OGB) and exhibit very low levels of heterophily.

The method and idea are very interesting, but the experiment does not sufficiently demonstrate their effectiveness and necessity. I will consider increasing the score once this issue is addressed.

**Questions:**

See weaknesses.

---

> ### Author Response · Authors · 2024-11-17
> **Rebuttal**
>
> Dear reviewer iy7r,
>
> Let us first thank you for the review and for appreciating the relevance of our work and our efforts in the experimental evaluation.
>
> **Weakness 1**
> We added a new Section F in the appendix dedicated to theoretical and empirical analysis of the complexity. The section includes:
> - Algorithmic complexity (we report the theoretical complexity using the big-O notation)
> - Execution times (we report the execution times of different pooling methods on the 5 datasets considered in the node classification task, which are the most demanding in terms of resources)
> - Memory usage (we report the utilization of the GPU memory on the node classification datasets and on a controlled experiment where we gradually increase the size of a synthetically generated graph and measure the memory consumption).
>
> Regarding the execution times and GPU utilization, we note that one-over-$K$ methods such as Graclus, NDP, and $k$-MIS perform a preprocessing step on the CPU before training starts.
> Those are not reported in the measurement, but they can take significant time and be a bottleneck in those cases that require operations such as eigenvalue decomposition.
>
> **Weakness 2**
> We created an anonymous version of our repository, which can be accessed at https://anonymous.4open.science/r/MaxCutPool/
>
> **Weakness 3**
> We agree that this is an important point and we added a clarification in section 3.2 of the revised paper.
>
> Rather than conflicting, the homophilic nature of traditional message passing is actually leveraged by our method. The basic idea is that after a standard homophilic message-passing, the stronger the association between a pair of nodes, the more they will end up containing the same information. Therefore, keeping both is redundant and one of them can be dropped following the strategy implemented by our pooling operator.
>
> In addition, we note that combining a homophilic network with a heterophilic pooling strategy provides extra flexibility to our model.
> As an example, let's consider the COLLAB dataset which is highly homophilic. The downstream task is able to guide the model even towards ignoring the auxiliary loss if needed.
>
> With that said, we showed that our model is particularly good in amplifying the node differences where present, making it perform particularly well in the case of heterophilic graphs.
>
> **Weakness 4**
> The "No pool" baseline is used to establish whether pooling -- in general -- can improve performance on a given dataset, which is not always the case. This is a rather standard procedure in graph pooling evaluations.
> For example, if the graphs are too small, using graph pooling can lower the performance w.r.t. the baseline.
> Nevertheless, even in those cases, it is still meaningful to compare different pooling operators to evaluate their capability of retaining enough information. That is why the ranking only focuses on those comparisons.
>
> We understand that the term "satisfactory" can be subjective. To address this, we have grounded our claims in objective measurements and rigorous statistical analyses. Our results consistently demonstrate that our method outperforms existing approaches, with statistically significant improvements across the evaluation metrics. In particular, statistical analysis (ANOVA followed by Tukey-HSD tests, both with p-value 0.05) revealed that our method is consistently equal or superior to the other pooling operators.
>
> **Weakness 5**
> Yes, the interpretation of MaxCutPool-NL is correct. However, the core concept of MaxCut is still there. In fact, the HetMP layers still provide the MAXCUT architectural bias that encourages dissimilarity between connected nodes.
> As shown by the experimental results, such architectural bias can be sufficient in some cases, even though the addition of an explicit optimization objective is often more effective than the sharpening operator alone.
>
> **Weakness 6**
> There seems to be a misinterpretation. The $h(\mathcal{D})$ is a measure of **homophily**, not heterophily, and GCB-H and MUTAG are strongly homophilic datasets. The advantage of MaxCutPool on heterophilic graphs is demonstrated by its superior performance on highly heterophilic datasets used in the node classification experiments and the Multipartite dataset.
>
> **Weakness 7**
> Also in this case, we note that $h(\mathcal{D})$ is a measure of homophily.
>
> Planetoid and OGB graphs are extremely homophilic:
>
> | Dataset   | Cora  | CiteSeer | PubMed | Arxiv | Proteins | Products |
> |-----------|-------|----------|--------|-------|----------|----------|
> | $h(\mathcal{D})$  | 0.664 | 0.766    | 0.627  | 0.655 | 0.794    | 0.808    |
>
> As such, to achieve high performance on those datasets it is sufficient to apply a few layers of homophilic MP rather than using the proposed node classification architecture with pooling layers.

---

> > ### Comment · Reviewer_iy7r · 2024-11-25
> >
> > Thanks for the detailed responses. Some of my concerns have been addressed, but the mechanism that MaxCutPool works on heterophilic graphs is still unclear (weakness 3). I will raise the score to 6.

---

> ### Author Response · Authors · 2024-11-26
>
> Thank you for your feedback and for raising the score.
>
> We built this figure (https://imgur.com/a/evolution-of-scores-node-selection-maxcutpool-Nf1MkMo) from our experiment's logs on the Multipartite dataset, which is completely heterophilic by design (each node is connected *only* to nodes of different colors).
> The plot shows how the HetMP layers in ScoreNet can assign similar scores to nodes that are completely disconnected, demonstrating how MaxCutPool effectively handles heterophilic structures.
>
> We note that the GNN is the same as used in our experiments with GIN layers. Clearly, using heterophilic MP layers would facilitate the learning even further and is a viable option in some special cases.
> However, our goal was to provide a general-purpose pooling layer that can be a drop-in replacement in standard GNNs, turning them into powerful models that can handle both homophilic and heterophilic graphs.

---

### Author Response · Authors · 2024-11-17
**General comment and summary of major changes**

We sincerely thank all reviewers for their thorough and constructive feedback.

We would like to clarify an important point that seems to have caused some confusion: MaxCutPool is designed as a drop-in replacement for pooling layers within GNNs with standard MP layers. Importantly, MaxCutPool deliberately *leverages*, rather than fights against, the properties of traditional message passing.

After several MP layers, connected nodes tend to have similar features due to oversmoothing. MaxCutPool uses this to its advantage by selecting from each local neighborhood a node as a representative, dropping others that have become redundant. The heterophilic components of MaxCutPool (HetMP layers in ScoreNet and MAXCUT auxiliary loss) make this selection process more effective by encouraging diverse sampling, but this happens within the context of a traditional homophilic message passing GNN.

This perspective helps explain several observations in our results: the method's effectiveness on both homophilic and heterophilic graphs, the ability to ignore the auxiliary loss when the task requires it, and the complementary rather than conflicting relationship between traditional MP layers and our heterophilic pooling.

We acknowledge that we could have made this clearer in the original submission and appreciate the opportunity to clarify it now. We have extensively revised the paper to better reflect this concept and addressed all the reviewers' concerns. We uploaded the revised manuscript with a version with highlighted changes attached at the end.

The major modifications to the paper include:

- A new subsection 3.2, clarifying the interplay between heterophilic and homophilic operations in our layer.
- A new Section F in the appendix containing the algorithmic complexity of our method, as well as experimental comparisons of time and memory usage of the various pooling layers.
- A publicly accessible, anonymous version of our repository, containing the code supporting our experiments. It is available at https://anonymous.4open.science/r/MaxCutPool/

---

### Meta-Review · Area_Chair_ooBs · 2024-12-20

**Metareview:**

**(a) Scientific Claims and Findings:**
The paper introduces a novel approach for computing the MAXCUT in attributed graphs—graphs with features associated with nodes and edges. This method is fully differentiable, enabling joint optimization of the MAXCUT alongside other objectives within a neural network framework. Leveraging the MAXCUT partition, the authors implement a hierarchical graph pooling layer for Graph Neural Networks (GNNs). This pooling layer is sparse, differentiable, and particularly effective for downstream tasks on heterophilic graphs, where connected nodes may have dissimilar features.

**(b) Strengths:**
* Innovative Pooling Mechanism: The introduction of a differentiable, feature-aware MAXCUT-based pooling mechanism addresses limitations in existing graph pooling methods, particularly in handling heterophilic graphs.
* Differentiability: The proposed approach's differentiable nature allows for seamless integration into end-to-end trainable GNN architectures, facilitating joint optimization with other network components.
* Applicability to Heterophilic Graphs: The method's suitability for heterophilic graphs expands the applicability of GNNs to a broader range of real-world scenarios where connected nodes may not share similar features.
* Computational Complexity: An analysis of the computational complexity of integrating a MAXCUT-based pooling mechanism into GNNs has been added to the appendix and is competitive with efficient pooling baselines.

**(c) Weaknesses:**
* Empirical Evaluation Scope: While the paper demonstrates the effectiveness of the proposed method, the empirical evaluations could be expanded to include a wider variety of datasets and comparisons with more baseline methods to comprehensively assess performance. Often, the pooling operation itself does not add much merit to solving a task. This leaves doubts on the relevance of pooling for many tasks.
* Empirical Results on Heterophilic Graphs: A deeper experimental analysis of the impact on heterophilic real world tasks would have been more supportive of the author’s claims. The baseline results in Table 4 are relatively low even in comparison with simple modifications either of the architecture or rewiring. The authors claim, the pooling layer is complementary to other approaches to improve GNN performance on heterophilic tasks, but do not support this claim by experiments. It could also be possible that other approaches are sufficiently strong and would not benefit from additional pooling.
* Theoretical Analysis: A deeper theoretical analysis of the proposed method's properties, such as stability and convergence, is lacking and could strengthen the paper's contributions.
* Computational Complexity: An experimental comparison of run-times with other methods could have been added to compare also the practical efficiency.

**(d) Reasons for Acceptance:**
After a thorough evaluation, I recommend accepting the paper based on the following considerations:
1. Novel Contribution: The paper presents a unique and innovative approach to graph pooling by introducing a differentiable, feature-aware MAXCUT mechanism, addressing a significant gap in current GNN methodologies.
2. Integration Capability: The differentiable nature of the proposed pooling layer allows for seamless incorporation into existing GNN architectures, facilitating end-to-end training and potential improvements in performance across various tasks.
While there are areas for improvement, such as expanding empirical evaluations, the paper's innovative approach and potential impact on the field justify its acceptance.

**Additional Comments On Reviewer Discussion:**

Most reviewers agree on the merit of the proposed work and suggest acceptance. Most of the criticism of Reviewer Unwn has been addressed during the rebuttal. However, they remain unconvinced about the comparability of pooling methods, a concern, which came up relatively late during the rebuttal.

A bigger concern from my perspective is the limited merit of the pooling layer on many GNN tasks. Many proposals to improve GNN performance on heterophilic tasks seem more impactful and empirical evidence is lacking that the proposed pooling approach would be impactful in combination beyond other more effective proposals.
However, the fact that MAXCUT can be optimised, the method is differentiable and scales (in principle) to large graphs make me lean towards accepting the paper.

---

### Decision · Program_Chairs · 2025-01-22

Accept (Poster)